# PPT-EVAL: A Benchmark for Computer-Use Agents on PowerPoint Tasks

Apurva Gandhi [* 1]  Vishwas Suryanarayanan [* 2]  Raja Hasnain Anwar [3 4]  Firoz Shaik [2]  Shubhang Desai [5]
Thong Q. Nguyen [6]  Muhammad Taqi Raza [3]  Vishal Chowdhary [2]  Graham Neubig [1]

## Abstract

Creating and editing slides is a rich, multimodal activity that is ubiquitous in professional and educational settings, making it an ideal testbed for real-world computer-use agents. Microsoft PowerPoint is among the most widely adopted and feature-rich environments for presentation creation. We introduce PPT-EVAL, a benchmark of 120 PowerPoint tasks across 12 files that cover both content creation and presentation editing scenarios, organized by difficulty. A central challenge in this domain is evaluation: tasks are complex, multimodal, and often admit many valid solutions. Moreover, today's agents frequently make only partial progress, which binary success metrics fail to capture. To address this, we design a robust evaluation framework to help create task-specific rubrics for PowerPoint tasks, taking inspiration from and building on past works for rubric-based evaluation. These rubrics award partial credit for intermediate steps, penalize unnecessary changes and poor aesthetics, and provide natural language feedback. This nuanced approach proves highly effective, achieving a Kendall's $\tau_b$ correlation of 0.77 with human judgments. We find that existing frontier agents still struggle with solving PowerPoint tasks, with strong models like Claude-4.5-Opus achieving only a 45% success rate and an average partial score of 57%. The benchmark repository is located at: https://github.com/microsoft/ppteval.

## 1. Introduction

Creating and editing presentation slides is a core activity underpinning communication across workplaces, classrooms, and conferences worldwide. Intelligent agents capable of assisting with or automating parts of this process could offer substantial productivity gains. According to Buffalo 7, 28.7% of business leaders report spending five or more hours per week creating slides, and employees devote over 10% of their working time to presentation preparation. Poor slide design and inefficiencies have tangible costs: 26% of respondents reported losing a potential customer and 25% an existing one due to inadequate presentation quality.

The intricacies of slide creation and editing also makes it an ideal testbed for computer-use agents. Effective agents must reason over and manipulate diverse multimodal content—including text, images, tables, charts, icons, layouts, transitions, and animations—to fully utilize modern presentation tools. Moreover, real-world presentation workflows are dominated by iterative editing rather than creation from scratch. Surveys indicate that most presenters reuse and adapt existing decks multiple times, and many organizations maintain shared "slide libraries" to facilitate remixing and reuse (Khoja, 2019; SlideUpLift Editorial Team, 2025).

Despite the importance of this domain, no existing benchmark captures the full complexity of realistic slide editing. Broad computer-use benchmarks touch many applications but lack depth in presentation software (Xie et al., 2024; Bonatti et al., 2025), while presentation-specific benchmarks either emphasize generation from scratch (Ge et al., 2025) or are restricted to tasks solvable through limited programmatic APIs (Guo et al., 2024), omitting native functionality such as design tools, advanced graphics, transitions, animations and more. As a result, realistic, GUI-level PowerPoint editing remains underexplored as a benchmark challenge.

To address this gap, we introduce PPT-EVAL, a benchmark for GUI-based interaction with the web version of PowerPoint (PowerPoint Online). Unlike API-bound benchmarks, PPT-EVAL enables agents to access the full functionality available to human users—including graphics, layouts, transitions, and animations—offering a realistic and comprehensive environment for assessing computer-use agents.

---

[*]Equal contribution  [1]Carnegie Mellon University [2]Microsoft [3]UMass Amherst [4]Work done at Microsoft internship [5]Work done at Microsoft; now at Google [6]Work done at Microsoft; now at Snowflake. Correspondence to: Apurva Gandhi <apurvag@cs.cmu.edu>, Vishwas Suryanarayanan <visuryan@microsoft.com>.

*Proceedings of the 43rd International Conference on Machine Learning*, Seoul, South Korea. PMLR 306, 2026. Copyright 2026 by the author(s).

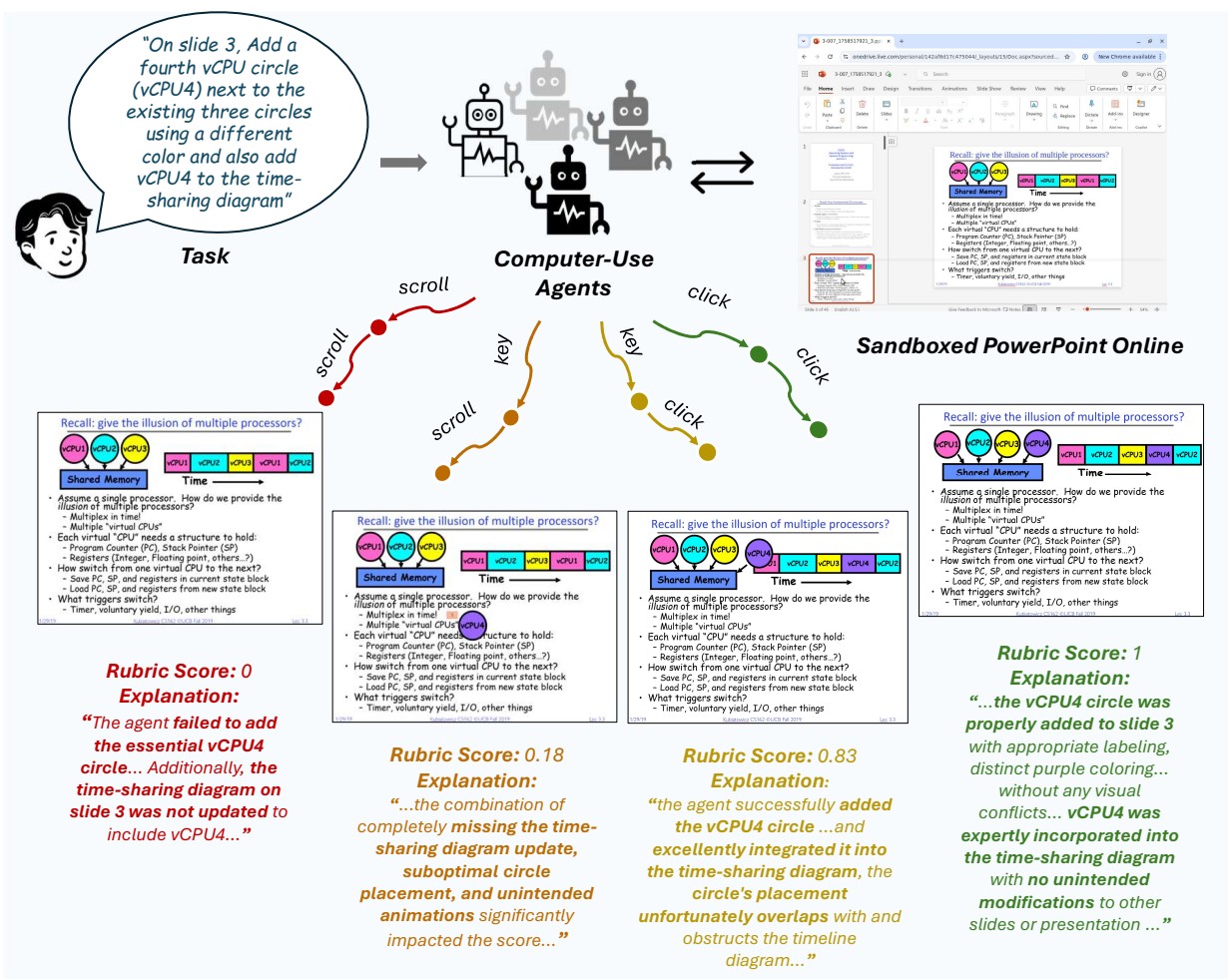

*Figure 1.* Illustration of task-solving in the PPT-EVAL Benchmark. Given a task, an agent interacts with a file in a sandboxed instance of PowerPoint Online. Interaction can take full advantage of PowerPoint's feature-set using the GUI. PPT-EVAL provides rubrics that can score different degrees of task completion and provide both nuanced partial credit and natural language feedback.

PPT-EVAL comprises 120 tasks sourced from openly licensed PowerPoint decks and stratified into easy, medium, and hard categories. Developing an evaluation framework for such tasks presents unique challenges: slide-editing goals are inherently multimodal, open-ended, and often admit multiple valid solutions. Given the immaturity of current computer-use agents, perfect task completion is challenging—agents can often only make partial progress. Binary success/failure metrics can therefore fail to capture meaningful distinctions in agent capability.

Inspired by prior work on rubric-based evaluation (Gou et al., 2025; Viswanathan et al., 2025), we design detailed rubrics for each task that (1) assign partial credit for intermediate progress, (2) penalize unnecessary or detrimental edits, and (3) generate natural-language feedback. This enables nuanced, interpretable scoring and robust automatic evaluation (see Fig. 1). In a meta-evaluation study, rubric-based scores show strong agreement with human judgments (Kendall's $\tau_b = 0.77$).

Finally, we evaluate a range of agents on PPT-EVAL, including proprietary frontier models such as OpenAI's COMPUTER-USE-PREVIEW (OpenAI, 2025) and Anthropic's CLAUDE-4-SONNET and CLAUDE-4.5-OPUS (Anthropic, 2025), as well as the open-weights models like the 7/8B and 32B variants of OPENCUA (Wang et al., 2025) and QWEN3-VL (Bai et al., 2025). We find that the strongest models can make meaningful progress (e.g., CLAUDE-4.5-OPUS achieves a 45% success rate and 0.57 average partial score) but still lag significantly behind human performance (80% success rate and 0.90 average partial score). Our findings highlight both the difficulty of the benchmark and the significant headroom for progress in realistic GUI-based computer-use capabilities.

## 2. Related Work

**Computer-use benchmarks.** OS-level benchmarks such as OSWORLD (Xie et al., 2024) and WINDOWSAGENTARENA (Bonatti et al., 2025) evaluate agents across a broad range of desktop applications in realistic environments either only offer superficial coverage of presentation software like LibreOffice Impress or omit such tasks entirely. OFFICEBENCH (Wang et al., 2024) targets multi-application office workflows involving Word, Excel, email, and calendar tools, yet excludes PowerPoint or other presentation focused tasks.

**Presentation-focused benchmarks.** PPTC (Guo et al., 2024) benchmarks PowerPoint editing through programmatic calls via `python-pptx` (Canny & contributors, 2025). This approach excludes agents from using features such as designer tools, advanced graphics and SmartArt support, themes, advanced layouts, transitions and animations, etc. Another benchmark, SLIDESBENCH (Ge et al., 2025), focuses on text-to-slide generation, assessing output similarity and design metrics for programmatically produced slides. It focuses on benchmarking single-slide creation from scratch rather than editing a whole slide deck within PowerPoint's full GUI environment, thus omitting the iterative, grounded editing typical of real workflows.

**Rubric and checklist-based structured evaluation.** While the benchmarks mentioned above use binary success/-fail criteria for measuring agent performance, recent work in web benchmarks and RL for non-verifiable domains, such as MIND2WEB 2 (Gou et al., 2025) and WILDCHECKLIST (Viswanathan et al., 2025), introduce structured rubrics or checklists that can capture degree of success. Another example is SHEETAGENT (Chen et al., 2024) which introduces a spreadsheet manipulation benchmark, pairing each task with a detailed sequence of subgoals that support partial-credit scoring. Inspired by these approaches, we design hierarchical, tree-structured rubrics for PPT-EVAL tasks that allocate partial credit, penalize extraneous edits, and generate natural-language feedback.

Table 8 in the Appendix summarizes the distinctions between PPT-EVAL and related benchmarks.

## 3. The PPT-EVAL Benchmark

### 3.1. PPT-EVAL Tasks

Each PPT-EVAL task consists of a **Goal** (a natural-language instruction), a **File** (the `.pptx` file to modify), and a **Rubric** (a structured scoring script). We design tasks that leverage PowerPoint Online's rich feature set. Because GUI agents can natively access these capabilities, while current API-based interaction typically supports only a subset, our main experiments focus on GUI-based agents. Nonetheless, the

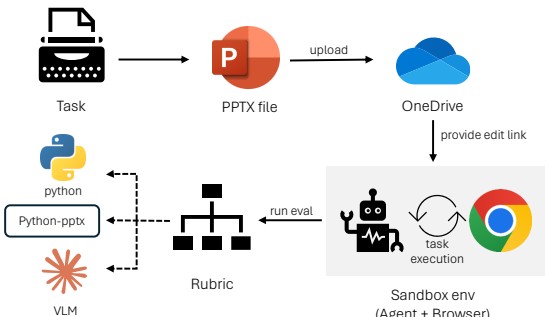

*Figure 2.* Task setup and evaluation workflow.

benchmark framework remains compatible with both approaches: evaluation depends solely on the *original* file and the agent's *modified* version—*not* on the sequence of actions taken. This design makes PPT-EVAL method-agnostic.

### 3.2. The PPT-EVAL Environment

Figure 2 depicts the PPT-EVAL task execution and evaluation workflow. To prepare a task, PPT-EVAL first uploads a copy of the task's PPTX file to OneDrive and obtains an *anonymous, editable* PowerPoint Online URL. The task is then launched inside an Ubuntu-based sandbox (instantiated with `screenenv` (Hugging Face, 2025)) running a Chromium browser pointed to this URL. This setup gives agents access to the full end-user feature surface of PowerPoint for the web, avoiding the coverage limitations of programmatic APIs such as `python-pptx`. Importantly, because the link provides anonymous edit access, the benchmark can be run *without* a Microsoft 365 subscription.[1]

Our environment provides interfaces for common GUI-level actions (mouse, keyboard, scrolling). The environment executes each action, advances the browser state, and returns a full-screen screenshot as the observation, closely mirroring human slide editing and capturing the multimodal nature of presentation manipulation. For reproducibility and parallelization, every run begins with a fresh copy of the file and a clean browser session, so tasks derived from the same deck remain independent and can be evaluated concurrently across multiple sandboxes.

### 3.3. File Selection

We selected a representative set of 12 openly licensed PowerPoint decks from the Internet Archive comprising 404 unique slides that provide a broad coverage of slide styles. Fig. 3 shows examples of slides across the files and Fig. 4

---

[1]Each task runs in an isolated PowerPoint Online session (via anonymous access), enabling deterministic initialization, safe execution, and parallel evaluation. The harness provides per-task timeouts, detailed logs, and artifact capture (before/after files and screenshots) to support systematic debugging.

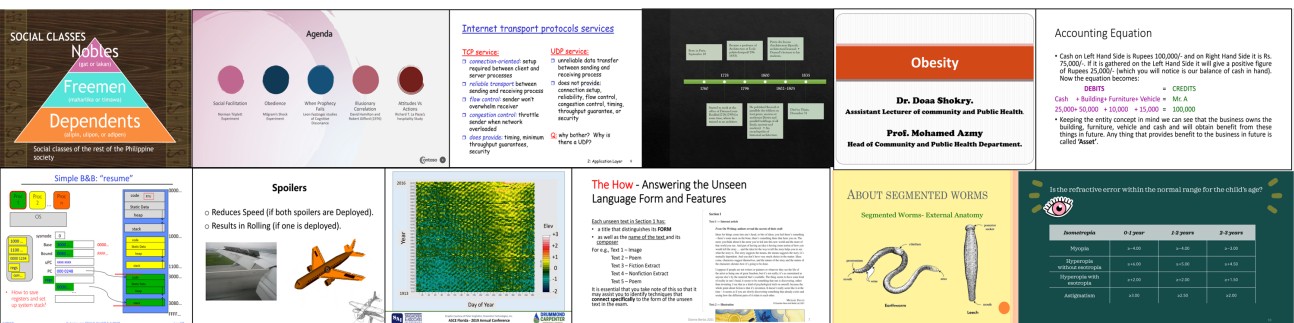

*Figure 3.* Representative slides from the 12 benchmark files, spanning topics (e.g., medicine, CS, accounting, history) and visual styles (text-heavy to graphics-heavy) to support heterogeneous task types.

shows the percentage of slides relevant to PPT-Eval tasks containing various elements (e.g., images, tables, animations, non-standard layouts, etc). The files span topics including Medicine, Computer Science, Accounting, Life Sciences, History, Aerospace, Architecture, Social Science, Education, and Environmental Science. File attributions and licenses are provided in Appendix A.1.

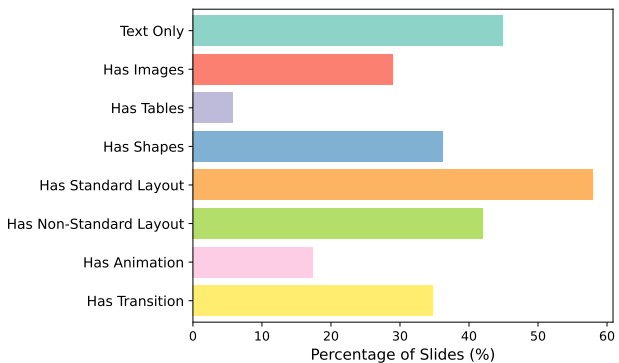

*Figure 4.* Distribution of elements in slides relevant to PPTArena's tasks. "Has Shapes" denotes slides containing shapes other than text boxes, images, or tables; "Standard Layout" denotes the default PowerPoint "Title and Content" layout.

### 3.4. Task Curation

We curated 10 tailored tasks for each of the 12 selected files (120 tasks total) using a semi-automatic pipeline. We started by generating a pool of 471 plausible task candidates across the files by prompting a computer-use agent (CLAUDE-4-SONNET) to explore each file and propose tasks. This process is inspired by recent work on using LLM-driven exploration to generate grounded computer-use tasks (Murty et al., 2024; Gandhi & Neubig, 2025; Zhao et al., 2025).

In particular, similar to Go-Browse (Gandhi & Neubig, 2025), we augmented the agent's action space with an additional tool (`add_tasks_to_dataset(tasks: list[str])`), which enables the agent to explicitly propose and log tasks during exploration. We ran this task-

proposal agent with a budget of 35 steps per file. Appendix A.2 provides the full task proposal prompt.

The resulting candidate tasks were then distributed among six human annotators who filtered the pool down to 10 final tasks per file and refined them for clarity, usefulness, and feasibility (Appendix A.3). Task difficulty was determined based on estimated user effort:

- **Easy:** Simple tasks that typically require $\leq 5$ steps or $\leq 1$ minute. These capture basic capabilities that are often easier for a human to perform directly. Nevertheless, these provide useful insight into agents' basic PPT proficiency.

- **Medium:** Slightly more complex tasks requiring about 5–10 steps or 2–5 minutes. These compound tasks are where a user begins to see real value agent delegation.

- **Hard:** Complex tasks that may take $\geq 10$ steps or $\geq 5$ minutes, requiring non-trivial reasoning or use of advanced features. Delegating such tasks to an agent would save users substantial time and effort.

Fig. 5 shows the task distribution by difficulty. Examples of tasks by difficulty levels can be found in Table 4 in the Appendix.

### 3.5. Task-Specific Rubric Design

Evaluating presentation-editing tasks is challenging because slide modifications are inherently multimodal and often admit multiple valid solutions. A successful evaluation scheme must verify not only whether required content appears, but also whether visual properties such as alignment, layout, and formatting match the intended outcome. At the same time, the evaluation must detect and penalize unintended changes, support partial credit for partially correct intermediate states, and remain agnostic to the agent's solving strategy—whether actions are produced through GUI control or through programmatic APIs.

To address these challenges, we build upon prior work on rubric-based structured evaluation for LLMs and agents (Gou et al., 2025; Viswanathan et al., 2025). Our design

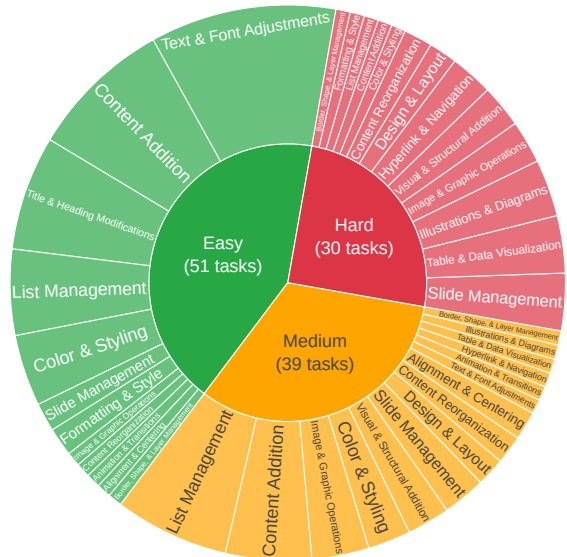

*Figure 5.* Distribution of the 120 tasks by difficulty (easy/medium/hard) and high-level intent categories (e.g., design & layout, image operations, table/data visualization). *Zoom to see intent categories.*

draws inspiration from the `RubricTree` framework of Mind2Web 2 (Gou et al., 2025), while introducing several key modifications to better suit the PowerPoint domain.

### 3.5.1. TREE-STRUCTURED RUBRICS

Each task is represented by a tree-structured rubric, where nodes correspond to evaluation criteria. Internal nodes define higher-level criteria, while leaf nodes implement concrete checks. Following Mind2Web 2, we distinguish between **critical** and **non-critical** criteria: *Critical nodes* correspond to core requirements of the task, necessary for meaningful progress. *Non-critical nodes* capture desirable but secondary aspects (e.g., stylistic choices, formatting consistency, penalizing extraneous changes, subjective, visual evaluation, etc.). An example rubric tree is illustrated in Fig. 6.

Each leaf implements a `compute_score() -> tuple[str, float]` function that returns both a numeric score ($\in [0,1]$) and a natural language explanation. These functions are executed in a global context populated with file paths to the original and modified presentations, along with screenshots of all slides. We also provide a custom `PPTDiff` class that supports common checks (e.g., whether unintended slides were modified, added, or removed) and extracts metadata such as animations and transitions from the XML representation—features not visible in screenshots or supported by `python-pptx`. Appendix A.6 shows examples of several leaf node implementations.

Finally, some criteria require assessing semantic or visual equivalence rather than exact matches. For example, a task may allow multiple reasonable ways to rephrase a slide title ("The King of the Jungle" vs. "The Great Cat"), or may require verifying that an inserted diagram preserves intended spatial relationships rather than pixel-perfect coordinates. To handle such cases, a leaf node's `compute_score` function may make use of an LLM call (to assess semantic correctness / relevance) or a VLM call (to assess correctness visually), as shown in Appendix A.6 examples.

### 3.5.2. AGGREGATION STRATEGY

Intuitively, we desire that our rubrics satisfy the following desiderata:

1) If an agent makes no progress, or only progress irrelevant to the task goals, the score is 0.

2) Meaningful intermediate checkpoints toward task success should receive partial credit, ideally proportional to the degree of completion.

3) If the task is completed perfectly, the score is 1.

Mind2Web 2 adopts a *gate-then-average* rule: if any critical child has a score of 0, the parent is forced to 0. Only if all critical nodes evaluate to 1, the parent node's score is set to the average of the non-critical nodes. We observe that this rule can often fail to reward partial progress in our setting. For instance, in the Fig. 6 example, the Mind2Web 2 strategy would assign a score of 0 even though the attempt made meaningful partial progress. We instead adopt a modified aggregation formula that yields better partial scoring as shown in the figure, and described below.

---

**Rubric scoring rules**

Let $s_i \in [0,1]$ denote the score of child $i$. Let $\mathcal{C}$ and $\mathcal{N}$ be the index sets of critical and non-critical children, respectively. Let us define the average child scores as:

$$\bar{s}_{\text{crit}} = \frac{1}{|\mathcal{C}|} \sum_{i \in \mathcal{C}} s_i, \qquad \bar{s}_{\text{non}} = \frac{1}{|\mathcal{N}|} \sum_{i \in \mathcal{N}} s_i.$$

- If both critical and non-critical children exist (i.e., $\mathcal{C} \neq \emptyset$ and $\mathcal{N} \neq \emptyset$),

$$\bar{s}_{\text{parent}} = \max\left\{0, \; \bar{s}_{\text{crit}} - \lambda\left(1 - \bar{s}_{\text{non}}\right)\right\}.$$

- Otherwise (all children are critical or all non-critical),

$$\bar{s}_{\text{parent}} = \frac{1}{|\mathcal{C} \cup \mathcal{N}|} \sum_{i \in \mathcal{C} \cup \mathcal{N}} s_i.$$

---

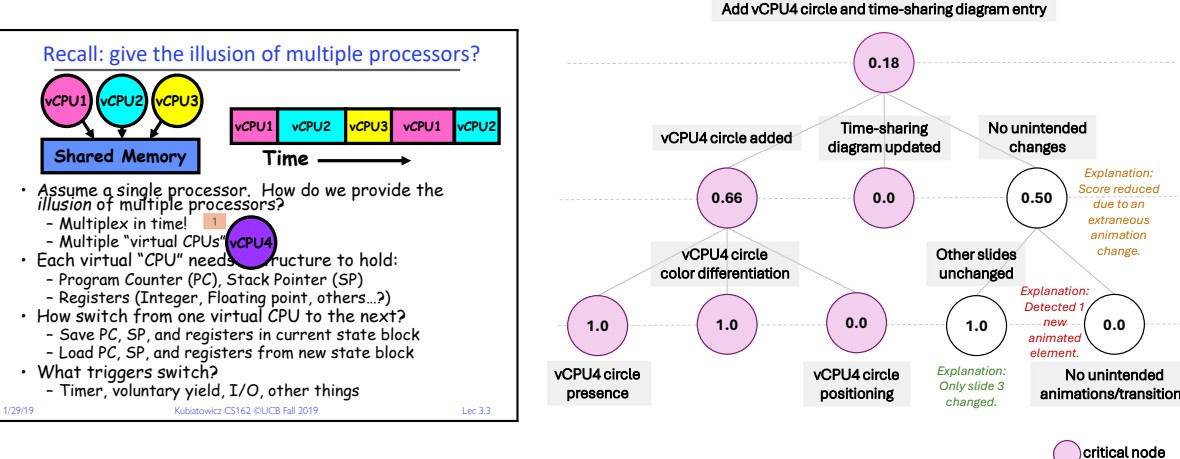

*Figure 6.* Example rubric tree and scores for the "add vCPU4" task from Fig 1: critical and non-critical check scores and explanations aggregate to a graded outcome with overall explanatory text. The rubric correctly identifies and accordingly awards partial progress a vCPU4 circle but not positioning it correctly nor updating the time-sharing diagram.

We set $\lambda = 0.3$ which controls the maximum penalty we can incur from failure on non-critical criteria. This formulation preserves the importance of critical checkpoints while still awarding proportional credit, and allows non-critical errors to introduce penalties without entirely zeroing out progress. In practice, this method better aligns with intuitive judgments of partial completion.

Similar, to score aggregation, leaf-level natural language score explanations are also recursively bubbled up. At each internal node, an LLM is prompted to synthesize a coherent explanation from the child scores and rationales. This produces interpretable feedback at every node of the rubric tree, culminating in a human-readable justification for the overall task score, as illustrated in Fig. 1 and 6.

### 3.5.3. RUBRIC CONSTRUCTION

Rubrics are generated semi-automatically. For each task, we first use LLMs (CLAUDE-4-SONNET and GPT-4.1) to propose a draft rubric tree, including both the structure and implementations of leaf scoring functions. While these drafts greatly accelerate rubric creation, they typically require human review and refinement.

We conducted two rounds of review with six human experts. In the first round, each annotator revised model-generated rubrics for tasks from two files. In the second round, rubrics were exchanged for cross-review and further polish. Annotators were instructed to simulate varying levels of task completion to test binary success/failure as well as partial credit behavior. Overall, rubric development required ∼150 hours of human effort. The prompt for rubric draft generation is provided in Appendix A.4 and specific examples of manual human effort required for rubric refinement are provided in Appendix A.5.

## 4. Experiments

### 4.1. Rubric Meta-Evaluation

To evaluate the reliability of our rubrics, we conducted a meta-evaluation study. We sampled 30 tasks from PPT-EVAL (2-3 per file) and asked our human annotators to create 2-4 solution attempts (PowerPoint files) per task, spanning various degrees of completion as described below:

1) **No Progress** (Expected Score 0): The attempt does not address any task requirements.

2) **Some Progress** (Expected Score $(0, 0.5)$): The attempt addresses some, but not all, requirements.

3) **Significant Progress** (Expected Score $[0.5, 1)$): Addresses most requirements with minor issues.

4) **Perfect Completion** (Expected Score 1): Fully satisfies all requirements.

We then used our rubrics to score each solution and compared the rubric scores against the expected score category. We reported category-wise accuracy as well as measured correlation by calculating the Kendall's $\tau_b$ correlation coefficient and Spearman's $\rho$ rank correlation coefficient. Note that after performing meta-evaluation, we further fixed issues we found with the sampled rubrics, before benchmarking the various agents. So, the meta-evaluation results slightly underestimate the final benchmark rubric quality.

## 4.2. Benchmarking Computer-Use Agents

We benchmark a range of computer-use models on PPT-EVAL. To represent closed, frontier models we use Anthropic's CLAUDE-4.5-OPUS and CLAUDE-4-SONNET as well as OpenAI's COMPUTER-USE-PREVIEW (CUA) models. In the open-weights categories we benchmark the 7/8B and 32B variants of the OPENCUA and QWEN3-VL-INSTRUCT models. Models are benchmarked with the native computer-use action space they were trained on, defined by the model providers. We use a maximum budget of 30 steps per task and set concurrency to 3 threads to speed up benchmarking (∼3.5 hours per run). For VLM and LLM calls in our rubrics, we use CLAUDE-4-SONNET.

We report two main success metrics: (1) **Success Rate (SR)** is the percentage of tasks that get a perfect score of 1, and (2) **Avg. Score** is the average of rubric scores across all tasks in a run, allowing us to consider partial scores.

## 4.3. Human Baseline

We also asked five human participants, separate from the 6 annotators who helped curate the benchmark, to attempt the benchmark tasks. These participants, come from a mix of backgrounds with varying PPT skill levels: QA, SWE, Marketing, Data Science professionals and a graduate student.

## 4.4. API Baseline

While this work primarily focuses on benchmarking GUI agents, we also add a strong API-based (or CLI) agent implemented using the Claude Code agent harness with the CLAUDE-4.5-OPUS model equipped with Anthropic's own `pptx` skill.

## 5. Analysis

### 5.1. Agent Performance

Overall, we find that today's GUI computer-use agents still struggle with PowerPoint tasks and significantly lag behind both human users and even behind strong API/CLI agents. Table 1 shows both aggregate results across all 120 tasks and categorized by difficulty level. The frontier proprietary models we benchmark, COMPUTER-USE-PREVIEW, CLAUDE-4-SONNET and CLAUDE-4.5-OPUS, achieve moderate scores on the benchmark (SR 0.39-0.45) with much headroom left compared to the human baseline scores (SR 80%). Smaller open-weights models perform noticeably worse with the best 32B model (OPENCUA-VL-32B) scoring a SR of 28%. The use of rubric-based partial scoring provides a more nuanced view of performance: even when success rates are close (e.g., the QWEN models), partial credit can help better understand performance differences. To further analyze performance, we also cat-

egorize each task based on PowerPoint structures that are most relevant to them (e.g., Shapes, Images, Tables, Animations, etc.) and plot model success rates with respect to this categorization in Fig 7. We discuss general patterns below and provide a further success-failure clustering analysis in Appendix E.1 as well as examples of agent trajectories in Appendix F.

**Comparison of frontier models.** In Table 1, we see that the CLAUDE models slightly outperform OpenAI's COMPUTER-USE-PREVIEW overall, with mostly higher average scores and success rates across all levels of difficulty. CLAUDE-4.5-OPUS outperforms CLAUDE-4-SONNET slightly overall, and especially on hard tasks where it has 18% lead over CLAUDE-4-SONNET. Though, CLAUDE-4-SONNET performs better on medium tasks. Fig. 7 helps explain some of finer-grained performance the differences between the models: while both CLAUDE-4-SONNET and CLAUDE-4.5-OPUS do well across categories, CLAUDE-4-SONNET performed especially well on tasks related to non-standard layouts and CLAUDE-4.5-OPUS performed better on tasks related to tables, shapes and text. Interestingly, COMPUTER-USE-PREVIEW performs much better on visual tasks such as those with shapes, tables and images, but struggles with layouts and animations.

**Comparison of Open-Weights models.** Among the open-weights models, we find that the OPENCUA models perform surprisingly well for the model sizes, with similar overall success rates (28% vs. 25%) for the 32B and 7B variants. While overall success rates are comparable, we see much more differentiation when looking at average partial scores: the 32B model often makes much more progress compared to the 7B model (0.45 vs. 0.34 avg. partial score). When looking across success rates by task categories in Fig 7, we see that the two OPENCUA model variants perform quite similarly across different categories except for text-related and animation-related tasks, where the 32B model performs better. The more general-purpose QWEN models perform noticeably worse than the computer-use-specialized OPENCUA models (14% and 13% SR for 8B and 32B respectively). Once again, we see a bigger difference in performance by looking at the avg. score (0.25 vs. 0.20). Interestingly, the 8B QWEN-3-VL-INSTRUCT model performs slightly better than the 32B model, but this result is consistent with their performances on OSWorld (Bai et al., 2025). In Fig 7, we see that while these models get some successes on tasks related to fundamental PPT structures (text, shapes and images), they perform a lot worse on tasks related to more advanced structures (tables, animations, non-standard layouts, etc.)

**Comparison to the API Baseline** Interestingly, even though the GUI agents have access to a richer feature-set via

*Table 1.* Performance on PPT-EVAL. Each cell reports *Success Rate (SR) / Avg. Score / Avg. Steps*. Results for closed models are averages over three runs. Additional variance analysis in Appendix B.5 and E.2.

| Model | Overall (120) | Easy (51) | Medium (39) | Hard (30) |
|---|---|---|---|---|
| Human Baseline | 0.80 / 0.90 / – | 0.88 / 0.92 / – | 0.78 / 0.89 / – | 0.68 / 0.88 / – |
| API Baseline (Claude-4.5-Opus) | 0.62 / 0.81 / – | 0.80 / 0.93 / – | 0.63 / 0.82 / – | 0.30 / 0.61 / – |
| *Closed Models.* | | | | |
| Claude−4.5−Opus | **0.45** / 0.57 / 20.96 | **0.56** / 0.67 / 16.83 | 0.38 / 0.52 / 22.86 | **0.35** / 0.48 / 25.52 |
| Claude−4−Sonnet | 0.42 / 0.53 / 12.31 | 0.55 / 0.61 / 9.87 | **0.42** / 0.57 / 12.74 | 0.17 / 0.33 / 15.92 |
| Computer−Use−Preview | 0.38 / 0.49 / 21.68 | 0.47 / 0.51 / 19.17 | 0.35 / 0.54 / 21.98 | 0.20 / 0.36 / 25.69 |
| *Open-weights Models.* | | | | |
| OpenCUA−32B | 0.28 / 0.45 / 15.89 | 0.35 / 0.51 / 13.98 | 0.33 / 0.50 / 13.67 | 0.10 / 0.29 / 22.03 |
| OpenCUA−7B | 0.25 / 0.34 / 19.86 | 0.31 / 0.33 / 17.53 | 0.26 / 0.39 / 20.05 | 0.13 / 0.29 / 23.57 |
| Qwen3−VL−8B | 0.14 / 0.25 / 21.23 | 0.20 / 0.32 / 17.92 | 0.15 / 0.22 / 22.03 | 0.03 / 0.19 / 25.83 |
| Qwen3−VL−32B | 0.13 / 0.20 / 23.82 | 0.20 / 0.27 / 23.20 | 0.10 / 0.14 / 22.49 | 0.07 / 0.17 / 26.60 |

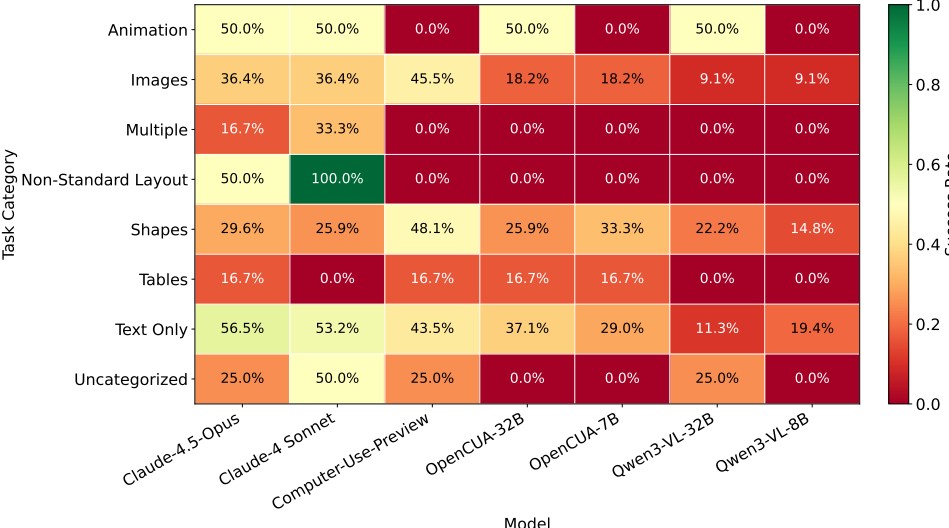

*Table 2.* Meta-evaluation results: per-category accuracies and overall correlation coefficients.

| Category | Accuracy |
|---|---|
| No Progress | 100% |
| Some Progress | 44.44% |
| Signif. Progress | 61.54% |
| Perfect Compl. | 88.89% |
| **Kendall's $\tau_b$** | 0.77 |
| **Spearman's $\rho$** | 0.84 |

*Figure 7.* Breakdown of model success rates with respect to task categories representing PowerPoint structures (Images, Shapes, Tables, Text, etc.) relevant to the task.

interaction with PPT Online, the API baseline still significantly outperforms the best GUI agent. This demonstrates the gap in maturity between current frontier training for CLI/API-based interaction vs. GUI-based interaction and the significant headroom remaining for improvement of GUI agents. While the API baseline excels at tasks that are easily solvable via current open-source programmatic APIs, it is unsurprisingly unable to solve tasks involving features that do not have good API coverage. For instance, the following are example tasks that at least one GUI agent could solve that the API baseline struggles to:

- Add a hyperlink on "HSC" linking to slide 1.

- Apply a different design theme to the entire presentation while maintaining the current color scheme of greenish backgrounds.

- Insert a SmartArt graphic on slide 3 representing the accounting equation (Assets = Capital + Liabilities) using an appropriate SmartArt style.

### 5.2. Rubric Accuracy

Table 2 reports both per-category accuracies and overall correlation coefficients from our meta-evaluation. The results indicate that rubric scores are strongly aligned with human annotator expectations. In particular, we observe a Kendall's $\tau_b$ correlation coefficient of 0.77 and a Spearman's $\rho$ rank correlation coefficient of 0.84, reflecting very strong consistency between rubric and human-assigned categories. Category-wise results further show perfect agreement for the *No Progress* class, high agreement for *Perfect Completion*, and moderate agreement for intermediate progress levels.

Double-clicking on some of the mismatches in rubric scores

and human expectations, we find that "perfect completion" mismatches were due to (1) human error (human accidentally making changes on the wrong slide and expecting a perfect score); (2) differing tolerance for subjective criteria (for instance reducing scores due to finding an emoji culturally insensitive); or (3) occasional VLM hallucinations. The partial credit categories ("some progress" and "significant progress") mismatches are often due to mix-ups where a human-labeled "some progress" attempt might be scored by the rubric as greater than 0.5 but less than 1, and vice versa. We provide some detailed case studies of rubric scoring in Appendices B.2 and B.3. We also provide a variance analysis for rubrics in Appendix B.5, demonstrating great stability of the rubrics across repeated runs.

## 6. Conclusion & Future Work

We introduced PPT-EVAL, an agent benchmark for PowerPoint creation and editing, that supports rich functionality through PowerPoint GUI support, with a rubric-based evaluation that (i) grants partial credit for meaningful progress, (ii) penalizes extraneous edits, and (iii) produces natural-language feedback. Across 120 tasks spanning three difficulty tiers and the full feature surface, rubric scores align strongly with human judgment (Kendall's $\tau_b$=0.77), enabling measurement beyond binary pass/fail. Frontier agents still struggle as complexity rises (e.g., CLAUDE-4.5-OPUS: SR/Avg. Score = 0.45/0.57), indicating substantial headroom for robust, general-purpose GUI agents.

We view several opportunities for future work. First, while our rubrics are strongly correlated with human judgement and supports partial grading, perfect partial grading that smoothly increases from a score of 0 to 1 is still challenging to achieve in a rich environment like PowerPoint. Second, while we use models to accelerate rubric creation, model generated drafts still require extensive human edits and revisions preventing us from scaling arbitrarily which would be useful in agent training scenarios (e.g., reinforcement learning). Finally, while our benchmark covers PowerPoint in depth, many user workflows are often multi-app, highlighting an opportunity for new cross-application benchmarks that take advantage of each applications full feature-set.

## Acknowledgement

We would like to thank Simran Khanuja, Wayne Chi and Yueqi Song for their thoughtful and insightful feedback on an earlier draft of this paper. Apurva Gandhi is supported by the Amazon AI PhD Fellowship.

## Impact Statement

This paper presents work whose goal is to advance understanding of the capabilities of computer-use agents in aiding with a common workflow: creating and editing presentations. We expect that this work can help users make more informed decisions in deciding how and when to incorporate agents in their presentation-building workflows.

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

xiv.org/abs/2505.10593.

## Appendix Contents

## A. Benchmark Curation

To build PPT-EVAL, we designed a semi-automatic pipeline that combines manual file collection, LLM-based task generation, and systematic human refinement. This process enabled us to capture a broad spectrum of realistic PowerPoint operations across presentations from diverse domains—including medicine, computer science, accounting, life sciences, history, aerospace, architecture, social science, education, and environmental science. The resulting benchmark tasks are carefully curated to remain feasible for agents, varied in scope and difficulty, and well-suited for consistent and robust evaluation.

### A.1. File Attributions

The PPT-EVAL benchmark dataset includes 12 PowerPoint presentations sourced from the Internet Archive under various Creative Commons licenses. All files are used in accordance with their respective license terms.

Eleven files are under Creative Commons Public Domain Mark 1.0 or CC0, requiring no attribution, but provided here for

transparency. One file requires explicit attribution under Creative Commons Attribution 4.0 International (CC BY 4.0).

**Required Attribution:** The file "application+layer+slide.pptx.pptx" is sourced from the Internet Archive (https://archive.org/details/application-layer-slide-pptx-on77) and is licensed under Creative Commons Attribution 4.0 International (CC BY 4.0). This work is used unmodified in our benchmark dataset.

All files are also attributed in Table 3.

*Table 3.* Source presentations used in the PPT-EVAL dataset with associated licenses and attributions.

| Filename | Creator | Internet Archive Page | License |
|---|---|---|---|
| Accounting Equation | Muhammad Mohsin | https://archive.org/details/accounting-equation | CC PDM 1.0 |
| Worms | Muhammed Abubakar Naseer | https://archive.org/details/GTAVC2005 | CC PDM 1.0 |
| 4._Pre-Colonial_Filipino_Culture.pptx | Almonissah Amirol | https://archive.org/details/AlmonissahArchives_20171223_1426 | CC PDM 1.0 |
| Aircraft_surface | Sachin Karbari | https://archive.org/details/AircraftSurface | CC PDM 1.0 |
| application+layer+slide.pptx | gg | https://archive.org/details/application-layer-slide-pptx-on77 | CC BY 4.0 |
| HSC Careers and Expo FINAL COM MOD.pptx | Dianne Berios | https://archive.org/details/humanexperiences_202105 | CC PDM 1.0 |
| 3 | Prof. John Kubiatowicz | https://archive.org/details/16_20210929 | CC0 |
| Obesity | osama mansour | https://archive.org/details/Obesity_201504 | CC PDM 1.0 |
| pediatic glasses when to prescribe1 | Dr Ahmed Elkomy | https://archive.org/details/pediatric-prep. | CC PDM 1.0 |
| Jean-Nicolas-Louis Durand | Mayank | https://archive.org/details/jeannicolaslouisdurand | CC PDM 1.0 |
| Revisiting Classical Social Experiments [Autosaved] | Syed Tabraiz Bukhari | https://archive.org/details/revisitingclassicalsocialexperiments | CC PDM 1.0 |
| Drummond_Troilo-Unseen Aspects of Sea Level Rise (final) | Rahman Davtalab | https://archive.org/details/drummondtroilounseenaspectsofsealevelrisefinal | CC PDM 1.0 |

**License Abbreviations:**

- CC PDM 1.0: Creative Commons Public Domain Mark 1.0;

- CC BY 4.0: Creative Commons Attribution 4.0 International

- CC0: Creative Commons Zero/Public Domain Dedication.

### A.2. Task Proposal

First, we generated a pool of 471 candidate tasks by prompting a computer-use agent powered by CLAUDE-4-SONNET to explore each file and propose plausible tasks to perform in each file, grounded in the content of the file. To support this, we extended the agent's action space with a add_tasks_to_dataset function, enabling it to explicitly log tasks during a 35-step exploration.

> **Task Proposal Prompt**
>
> Explore the current file and propose tasks to add to the dataset.
> When adding tasks to the dataset, tag each task as easy, medium, or hard.
> Also include a slide number for each task. You can do this by using the function add_tasks_to_dataset with a list of tasks and using the [DIFFICULTY][SLIDE:N] <task> format for each task you add to the dataset.
>
> Make sure to include a variety of tasks that real users would want to do. Though, the benchmark will evaluate computer use agents instead of actual people, so do not propose tasks that require personal information.
>
> In our benchmark, we will be creating reward functions for each task using a mix of programmatic validation in python using python-pptx) and LLMs/VLMs with slide screenshot access (only slides, not the GUI, notes, comments, etc.) and some animation/transition validation. Please make sure to limit proposed tasks to ones that can be evaluated automatically in such a manner. E.g., do not propose tasks related video, audio, etc. that cannot be evaluated with just the above context.

> Note that there may be more slides than what you can see in the current view, so you may need to scroll to see all slides. You can add tasks as you are exploring the file, instead of waiting until the very end.
>
> When you are done, call the function `finish` with a message to the user with a reason for finishing.

*Table 4.* Example PPT-EVAL tasks by difficulty, illustrating typical edit operations from simple formatting to multi-step layout and content changes.

| Difficulty | Task |
|---|---|
| Easy | On slide 1, add a new text box below the author name that says 'An Introduction to Flight Dynamics'. |
|  | Change the background color of slide 3 to light blue. |
|  | Center align the title text on slide 4. |
| Medium | Sort the table data on slide 10 in alphabetical order by isometropia type. |
|  | On slide 8, add a new process box 'Proc 3' in purple color next to the existing processes. |
|  | Replace the background image of slide 2 with a lighter one. |
| Hard | Change the slide layout to "Three Content" and apply it to slide 14, rearranging content into a three-column comparison format, adding a new column that calls out the difference between unilateral and bilateral. |
|  | On slide 11, replace the existing diagram with a simpler flowchart showing 'Browser → HTTP Request → Web Server → HTTP Response → Browser'. |
|  | Change the color tone of the figure on slide 4 to blue by adding a shape on top of the figure and increasing its transparency. |

## A.3. Task Selection (Human Effort)

The 471 tasks generated by the model by exploring files provide a good set of candidates that are grounded in the content of the files. To ensure that we include a high-quality and diverse set of tasks in the benchmark and to keep it from being too computationally expensive to run, six human annotators filtered the task candidates to 10 tasks per file (120 in total) and then further refine or modify the tasks to ensure quality. The instructions for the task filtering step are provided below: annotators were asked to filter tasks based on the feasibility of solving them, the feasibility of evaluating them automatically, and to ensure diversity in task difficulty according to the definitions provided below.

> **Task Selection Instructions**
>
> For each of your assigned files, we want to choose 10 tasks and generate graders/rubrics for them.
> Choose tasks based on the following criteria:
>
> 1. **Feasibility of the task**
>
> 2. **Feasibility of automatic grading/evaluation**
>
> 3. **Diversity** – Try to avoid repetition of same/similar styles of tasks and choose tasks spread across the presentation rather than concentrated on the same slides.
>
> 4. You may need to **rephrase certain tasks** that are phrased ambiguously.
>    - Task rephrasing has a large impact on task difficulty and rubric generation.
>
> 5. Aim to select **3 easy, 4 medium, and 3 hard** tasks per file:
>    (a) **Easy**: Simple commanding tasks (∼requiring ≤ 5 steps; ≤ 1 min). e.g.,
>       i. Insert a subtitle below the title saying "..." on slide 5.
>       ii. Change the background color of slide 5 to sky blue.
>    (b) **Medium**: Slightly more complex (∼2–5 min; 5–10 steps), compound tasks where a user will likely start seeing value in delegating these to an agent. e.g.,
>       i. Animate the bullets on slides 3, 4, and 6, so that they show up one-by-one.
>       ii. Replace the background image on the title slide with a different image of a clock.
>    (c) **Hard**: These are complex tasks that may take the average user a non-trivial amount of time (≥ 5 min; 10+ steps) to perform/figure out themselves. e.g.,

> i. Create a summary slide right before the Thank You slide that adds a table summarizing the characteristics and differences of each of the worms.
> ii. Create a timeline graphic based on the "History" section with the key dates annotated.

After filtering down to 120 task candidates, the tasks were further refined and rephrased by the annotators. More than 31% of the 120 tasks were further rephrased. We asked annotators to note down the reason for rephrasing when they did perform one. We cluster these reasons into high-level categories and plot it in Fig. 8a. Below are some concrete examples of rephrasing tasks:

| Original | Rephrased | Reason |
| --- | --- | --- |
| **Improve Clarity / Specificity** | | |
| On slide 11, change the yellow ~~highlighted~~ text ~~to use~~ blue highlighting ~~instead~~. | On slide 11, change the yellow font color text to blue highlighting. | Text was using font color rather than highlight — rewording clarifies the change. |
| Add a new team member ~~to the team members slide~~ with the name 'Dr. Sarah Johnson' and role 'Pediatric Ophthalmologist'. | Add a new team member on the first slide with the name 'Dr. Sarah Johnson' and role 'Pediatric Ophthalmologist'. | There is no dedicated team members slide; names appear on the first slide. |
| **Increase Difficulty / Scope** | | |
| Sort the table data on slide 10 ~~in ascending order based on the first column values~~. | Sort the table data on slide 10 alphabetically by isometropia type. | Increases complexity of the task and requires interpretation before acting. |
| Create a callout box ~~around the entire second checkbox item about "NO RELATED TEXT"~~. | Create a callout box with no fill around the second checkbox item on slide 5. | Forces the model to determine which item is second. |
| **Rephrasing Infeasible Tasks** | | |
| Change the lecture number from 'Lecture 3' to 'Lecture 1' ~~both in the title and in the slide footers. The footer is in the slide master.~~ | Change the lecture number from 'Lecture 3' to 'Lecture 1' in the presentation title. | Slide master changes cannot be performed in PPT Online; must be done manually. |
| Change the color tone of the figure on slide 4 ~~so that its background looks light blue~~. | Change the color tone of the figure on slide 4 to blue by overlaying a shape and increasing transparency. | Figure color cannot be changed directly in PPT Online; workaround required. |
| **Correcting Inaccuracies** | | |
| Change the title ~~from 'Texts and Human Experiences' to 'Literature and Human Experiences'~~. | Change the title to 'Literature and Human Experiences - The Common Modules'. | The original title was not accurate to the intended content. |
| ~~Change the font size of the question labels.~~ | Adjust the question label font size for consistency. | The existing font size was already larger than 18pt and required uniformity. |

*Table 5.* Examples of manual task refinement.

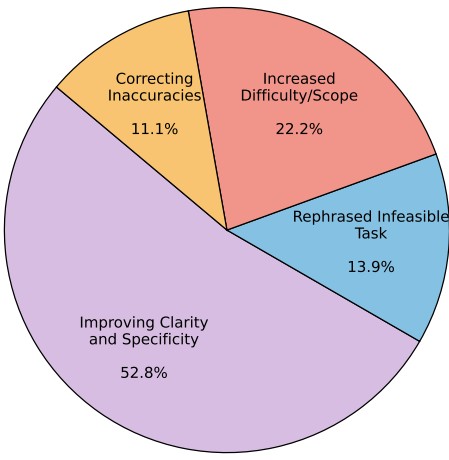

*(a)* Task Refinement Reason Distribution.

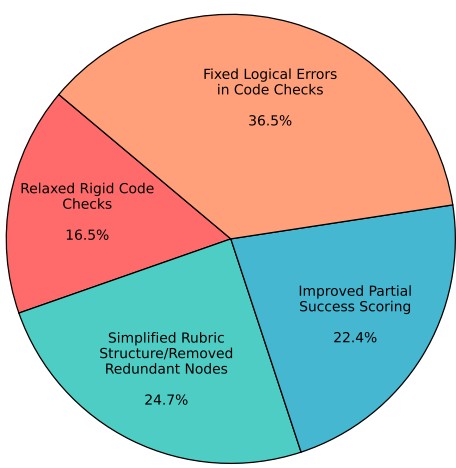

*(b)* Rubric Modification Reason Distribution.

*Figure 8.* Distribution of types of manual task (left) and rubric (right) refinement performed by Human Annotators.

### A.4. Rubric Draft Generation

For each task in PPT-EVAL, we first prompt an LLM (e.g., `Claude-4-Sonnet` or `GPT-4.1`) to generate a draft rubric that decomposes the task into a tree of evaluation criteria. The rubric specifies both critical requirements and non-critical aspects, ensuring that core goals are enforced while still awarding partial credit for meaningful progress.

Each leaf in the rubric is implemented as a check: programmatic checks use `python-pptx` and a custom `PPTDiff` class to analyze structure, content, animations, and transitions, while LLM/VLM calls handle semantic or visual judgments such as verifying formatting, relevance, or color. In all cases, rubrics also penalize extraneous edits and return both a score in $[0, 1]$ and a natural language explanation, providing a nuanced and interpretable evaluation of agent performance.

While the model-generated drafts provide a starting point for task rubrics, we found that these require significant refinement and modification by human experts in order to ensure high evaluation accuracy. We discuss this manual rubric refinement stage next in Appendix A.5.

---

## Rubric Generation Prompt

We are building a rubric to evaluate a task. We will do this by decomposing success criteria for the task into a rubric tree. The rubric tree should comprehensively test that the task is successfully completed and also penalize extraneous behavior.

In particular:

1. A rubric tree consists of nodes that each refer to a particular criterion.

2. A criterion can be decomposed into sub-criteria and so on.

3. A criterion node can be critical or non-critical.

4. A parent node's score computation depends on whether its children are critical, non-critical, or a mix of both.

5. If both critical and non-critical children exist, the parent score is

$$\max(0, \text{average(critical)} - \lambda \times (1 - \text{average(non-critical)})),$$

   where $\lambda =$ '%.2f' $|$ format(non_critical_weight) .

6. Otherwise (all children critical or all non-critical), the parent score is the average of all children.

7. A leaf node's score is computed using a particular scoring script written for that leaf node.

The rubric tree should be as comprehensive as possible, and should be able to evaluate the task in a way that is fair and accurate.
The rubric tree should be as concise as possible, and should be able to be easily understood by a human. The rubric tree should be as easy to evaluate as possible.

We are currently developing a benchmark to evaluate Computer-Use Agents in Microsoft PowerPoint. As part of this we are generating a rubric tree of criteria to evaluate whether the agent was successful in performing a task.

Here are some imports and class definitions that leaf node scorer functions will have access to:

```python
@dataclass
class AnimationEffect:
    '''Represents a single animation effect'''

    slide_id: str
    element_id: str
    effect_type: str
    trigger: str
    delay: float
    duration: float
    order: int
    # Note, element_text may sometimes be a superset of the finer grained text
    actually animated.
    element_text: Optional[str] = None
    element_type: Optional[str] = None

    def to_dict(self) -> Dict[str, Any]:
        ...
```

```python
@dataclass
class SlideTransition:
    '''Represents a slide transition'''

    slide_id: str
    transition_type: str
    duration: float
```

```python
        direction: Optional[str] = None

    def to_dict(self) -> Dict[str, Any]:
        ...
```

```python
@dataclass
class Slide:
    '''Represents a slide with its metadata'''

    slide_id: str
    slide_number: int
    title: Optional[str] = None
    layout_type: Optional[str] = None
    element_count: int = 0
    notes: Optional[str] = None
    content_hash: Optional[str] = None

    def to_dict(self) -> Dict[str, Any]:
        ...
```

```python
@dataclass
class PowerPointDiff:
    '''Container for PowerPoint differences'''

    added_animations: List[AnimationEffect]
    removed_animations: List[AnimationEffect]
    modified_animations: List[Tuple[AnimationEffect, AnimationEffect]]
    added_transitions: List[SlideTransition]
    removed_transitions: List[SlideTransition]
    modified_transitions: List[Tuple[SlideTransition, SlideTransition]]
    added_slides: List[Slide]
    removed_slides: List[Slide]
    modified_slides: List[Tuple[Slide, Slide]]

    def to_dict(self) -> Dict[str, Any]:
        ...
```

```python
@dataclass
class SlideScreenshot:
    '''Represents a slide screenshot'''
    slide_number: int
    image_path: str
    slide_id: Optional[str] = None
```

Besides these, the following packages are also installed and you may import them: `python-pptx`.
Scorer functions will also have access to the following global variables:

```python
ppt_diff: PowerPointDiff
original_ppt_screenshots: List[SlideScreenshot]
modified_ppt_screenshots: List[SlideScreenshot]
original_ppt_path: str
modified_ppt_path: str
```

Scorer functions will also have access to the following functions:

```python
def llm_call(prompt: str, temperature: float = 0.7, max_tokens: int | None = None)
    -> str:
    '''Call the LLM client with the given prompt.

    Args:
        prompt: The prompt to send to the LLM.
```

```
        temperature: The temperature to use for the LLM.
        max_tokens: The maximum number of tokens to generate.

    Returns:
        The response from the LLM.'''
```

```python
def vlm_call(prompt: str, images: List[Union[str, bytes]], temperature: float =
    0.7, max_tokens: int | None = None) -> str:
    '''Call the Vision LM client with the given prompt and images.

    Args:
        prompt: The prompt to send to the VLM.
        images: The images to send to the VLM. Each image can be:
            - File path (string) - will be read and base64 encoded
            - Base64 encoded string
            - Raw bytes - will be base64 encoded
        temperature: The temperature to use for the VLM.
        max_tokens: The maximum number of tokens to generate.

    Returns:
        The response from the VLM.'''
```

Please generate a comprehensive rubric tree for the following task.

Task: {{task}}

Return the rubric as a JSON structure with the following format:

```json
{
    'name': 'Root criterion name'
    'description': 'Detailed description of what this criterion evaluates',
    'is_critical': true/false,
    'children': [
        {
            'name : 'Child criterion name',
            'description': 'Description',
            'is_critical': true/false,
            'children': [...] // or 'scorer' for leaf nodes
        }
    ]
}
```

For leaf nodes, instead of `"children"`, include one of the following formats: {{scorer_formats}}.

Make sure the rubric is comprehensive, follows the scoring rules described above, and has appropriate critical/non-critical designations.

Function Scorer:

```json
{
    'type': 'function',
    'function_code': 'def compute_score() -> tuple[str, float]:\n    ...\n
    return \'<REASON_FOR_SCORE>\', <SCORE> # The score should be between 0 and 1.\n'
}
```

LLM Scorer:

```json
{
    'type': 'llm',
    'system_prompt': '...',
    'user_prompt': '<DESCRIPTION OF THE TASK TO EVALUATE> ... <INCLUDE ANY CONTEXT
    WITH VARIABLES USING JINJA2 TEMPLATE STYLE> ... Respond with JSON in a code
    block with score between 0 and 1: {\'reason\': \'..\', \'score\': X.XX}\n'
}
```

## A.5. Rubric Refinement (Human Effort)

The model generated rubric drafts for tasks were distributed between six human experts who put significant effort in refining and fixing problematic rubrics. In fact, we found that more than 81% of the model-generated drafts had to be fixed. When performing a modification to a rubric, the experts were asked to note down a summary of the modifications they made. We cluster these into high-level categories and plot the distribution in Fig. 8b. Below we provide some concrete examples of modifications made for each category:

*Table 6.* Manual rubric modification examples.

| Task ID | Goal | Issue | Fix |
|---|---|---|---|
| **Logical Errors** | | | |
| pediatic glasses when to prescribe1-035 | Change the slide layout to "Three Content" and apply it to slide 14, rearranging content into a three-column comparison format, adding a new column that calls out the difference between unilateral and bilateral | Generated rubric checked if the slide's layout was set to Three Content, but did not check if the content was reorganized into 3 columns, and the new column called out the difference between unilateral and bi-lateral amblyopia. | Two additional nodes were added that checked whether the content was redistributed into 3 columns, and the third column contained relevant content about the difference between unilateral and bilateral amblyopia. |
| pediatic glasses when to prescribe1-037 | Add a text box to slide 1 with the text "More Information", and then add a hyperlink on the text 'More Information' that links to slide 34 | The LLM generated a node to check if the text box was added, with the correct text, but generated flawed code for checking the existence of the hyperlink. | The code was re-written to use the representation of targets for links in presentation files buried deep in the XML and needs to be extracted via python-pptx's relationships APIs. |
| **Improving Partial Credit** | | | |
| Drummond_Troilo-Unseen Aspects of Sea Level Rise (final)-019 | On slide 5, Remove the '43 year horizon' red annotation from the chart | The generated node checks if the 43 year horizon annotation has been removed from the chart, but does so in a monolithic way – the task is all or nothing. | The check was further broken down into two components: (1) was the red text that says "43 year horizon" removed from the chart? (2) was the arrow that points to the portion of the chart removed? This allowed us to ensure that the model receives partial credit. |
| Accounting Equation-004 | Replace the word 'Liability' with 'Debt' throughout the slides | The generated code only checked that the string 'Liability' was no longer in the slides. While correct, it provided no way to provide partial credit. | The node was changed to count the number of instances of the string in both the original and final presentation, the proportion of replaced instances was used to provide the final score. If no instances were found, the node would return a perfect score, otherwise it would return a fractional score. |
| **Relaxing Rigid Rubrics** | | | |
| Obesity-011 | Convert the first bullet in the content placeholder for Slide 9 into three: (1) The association may be stronger for obese adolescents than younger children (2) Obese children are also more likely to have increased risk of heart disease (3) Obese children are also more likely to develop asthma, then change only the first bullet point to a numbered list item '1' on slide 9. | Generated code tries to check whether bullets are available in the text, but hallucinates how bullets are exposed python-pptx. Additionally, there are multiple methods to surface bullets in PowerPoint, and code-based checks are not exhaustive. | Fixed the code to use a VLM to check if the text is bulleted and has the correct content instead, making this more robust. |

| Worms-003 | Change the background color of Slide 4 to light blue | The generated code tries to use numpy to verify the color. This results in a very fragile check for the correct color (blue). | This is a problem uniquely suited for VLMs to solve, and changing to a VLM call makes this task much more robust and repeatable. |
|---|---|---|---|
| **Simplifying Rubrics** | | | |
| 4._Pre-Colonial_Filipino_Culture-005 | Combine slides 2 and 3 into a single slide with table for male and female clothing. Include the images from both slides. Don't delete the original slides yet; just insert this as a new slide after slide 3. | Generated rubrics were complicated, in that they checked for changes on all slides (besides 2 and 3). Additionally, the model attempted to make use of python-pptx to check for unnecessary changes instead of PPTDiff, which resulted in a complex subtree for extraneous change check (one per slide). While more branching is ideal for critical nodes, for non-critical nodes, this is mostly just noise. | Redundant nodes were removed and unnecessary change detection was simplified using PPTDiff instead of per-slide VLM checks. |
| Obesity-004 | Add a bullet point list with three items: 'Obesity definition', 'Global statistics', and 'Impact on health' as a new slide between 1 and 2, titled "Agenda" | The LLM added a redundant node checking for the slide position, and also generated checks for unnecessary changes by asking a VLM to compare the before and after for every single slide. | Redundant nodes were removed and the VLM based check was replaced by a PPTDiff check. |

## A.6. Example rubric leaf nodes

Below are some example leaf nodes in PPT-Eval:

## 1. Node using VLM:

```python
def compute_score() -> tuple[str, float]:
    # Compare slide 11 screenshots to detect color changes from yellow to blue
    original_slide_11 = None
    modified_slide_11 = None

    for screenshot in original_ppt_screenshots:
        if screenshot.slide_number == 11:
            original_slide_11 = screenshot
            break

    for screenshot in modified_ppt_screenshots:
        if screenshot.slide_number == 11:
            modified_slide_11 = screenshot
            break

    if not original_slide_11 or not modified_slide_11:
        return 'Could not find slide 11 screenshots for comparison', 0.0

    prompt = '''Compare these two PowerPoint slide images (before and after).

    Look specifically for:
    1. Yellow color text in the original image
    2. Whether any of that yellow color text has been highlighted with blue
    highlighting in the modified image

    Respond with:
    - \'YES\' if you can identify at least some yellow color text that has been
```

```
    changed to blue highlighting
    - \'NO\' if no yellow color text has been changed to blue highlighting
    - 'UNCLEAR' if you cannot clearly determine the highlighting colors or changes

    Focus only on highlighting colors (background colors behind text), not text colors
    themselves.'''

    response = vlm_call(prompt, [original_slide_11.image_path,
    modified_slide_11.image_path], temperature=0.1)

    if 'YES' in response:
        return  At least some yellow colored text has been changed to blue
    highlighting', 1.0
    elif 'NO' in response:
        return 'No yellow colored text has been changed to blue highlighting , 0.0
    else:
        return f'Unclear result from visual analysis: {response}', 0.5
```

## 2. Node using `python-pptx`:

```python
def compute_score() -> tuple[str, float]:
    '''Check that the (only) image on slide 6 is rotated ~45 degrees clockwise.'''
    from pptx import Presentation

    SLIDE_IDX = 5  # slide numbers are 1-based
    TARGET_ROTATION = 45
    ROTATION_TOLERANCE = 2  # degrees

    # Load modified presentation
    prs = Presentation(modified_ppt_path)
    if len(prs.slides) <= SLIDE_IDX:
        return ('Slide 6 does not exist in modified presentation.', 0.0)
    slide = prs.slides[SLIDE_IDX]

    # Collect debug info about shapes
    shapes_debug = []
    picture_like_indices = []
    pics = []
    for idx, sh in enumerate(slide.shapes):
        stype = getattr(sh, 'shape_type', None)
        name = getattr(sh, 'name', '<no-name>')
        fill_type = None
        try:
            fill = getattr(sh, 'fill', None)
            if fill is not None:
                fill_type = getattr(fill, 'type', None)
        except Exception:
            pass
        has_image_attr = hasattr(sh, 'image')
        # Primary detection: native picture shape_type == 13
        if stype == 13:
            pics.append(sh)
            picture_like_indices.append(idx)
        # Secondary detection: non-picture shape that has a picture fill
        elif has_image_attr or (fill_type is not None and
    str(fill_type).upper().endswith('PICTURE')):
            pics.append(sh)
            picture_like_indices.append(idx)
        shapes_debug.append({
```

```python
            'idx': idx,
            'name': name,
            'shape_type': stype,
            'has_image_attr': has_image_attr,
            'fill_type': str(fill_type) if fill_type is not None else None,
        })

    if not pics:
        debug_msg = (
            f'No image found on slide 6. Total shapes={len(slide.shapes)}. '
            f'Shape summaries: '
            + '; '.join(
                f'#{d['idx']} type={d['shape_type']} name='{d['name']}'
    has_image={d['has_image_attr']} fill_type={d['fill_type']}'  # noqa: E501
                for d in shapes_debug
            )
        )
        return (debug_msg, 0.0)

    # If multiple, pick first but include note
    pic = pics[0]
    if len(pics) > 1:
        multi_note = f' (Multiple picture-like shapes detected
    indices={picture_like_indices}; using first index {picture_like_indices[0]})'
    else:
        multi_note = ''

    rotation = getattr(pic,  rotation , 0) % 360
    diff = min(abs(rotation - TARGET_ROTATION), abs(rotation + 360 - TARGET_ROTATION))

    if diff <= ROTATION_TOLERANCE:
        return (f'Image rotation {rotation} degrees within tolerance.{multi_note}',
    1.0)
    return (f'Image rotation {rotation} degrees; expected {TARGET_ROTATION} +/-
    {ROTATION_TOLERANCE} degrees.' + multi_note, 0.0)
```

### 3. Node using `PPTDiff`:

```python
def compute_score():
    # Checks that all text on the references slide has 'Fly In' animation from the
    left.
    slide_ids = set()
    for slide in ppt_diff.added_slides + [s2 for _, s2 in ppt_diff.modified_slides]:
        if slide.title and 'references' in slide.title.lower():
            slide_ids.add(slide.slide_id)
    if not slide_ids:
        from pptx import Presentation
        try:
            pres = Presentation(modified_ppt_path)
            for i, slide in enumerate(pres.slides):
                for sh in slide.shapes:
                    if sh.has_text_frame and 'references' in sh.text.lower():
                        slide_ids.add(slide.slide_id)
        except Exception:
            pass
    if not slide_ids:
        return 'No references slide id found. Cannot check animations.', 0.0
    # Find all text elements on references slide(s)
    from pptx import Presentation
    pres = Presentation(modified_ppt_path)
```

```python
    text_elements = []
    references_slide_numbers = []
    for i, slide in enumerate(pres.slides):
        if hasattr(slide, 'slide_id') and slide.slide_id in slide_ids:
            references_slide_numbers.append(i+1)
            for sh in slide.shapes:
                if sh.has_text_frame and sh.text.strip():
                    text_elements.append(sh.text.strip())
    # If no text found, fail
    if not text_elements:
        return 'No text found on references slide to animate.', 0.0
    # For each text element, check if an appropriate animation was applied
    # We'll match by slide_id and try to match text (if available)
    animated_texts = []
    for anim in ppt_diff.added_animations + [a2 for _, a2 in
ppt_diff.modified_animations]:
        if anim.slide_id in slide_ids and anim.effect_type.lower() == 'fly in' and
anim.trigger == 'on click' and anim.element_type == 'text' and (anim.element_text
is not None):
            if anim.element_text.strip() in text_elements:
                if anim.direction and anim.direction.lower() == 'from left':
                    animated_texts.append(anim.element_text.strip())
    # Score: proportion of reference slide text that was animated correctly
    matched = set(animated_texts)
    unmatched = set(text_elements) - matched
    if not matched:
        return f'No references text was animated with Fly In from left.', 0.0
    if not unmatched:
        return 'All references text correctly animated with Fly In from left.', 1.0
    partial = len(matched) / max(1, len(text_elements))
    return f'Some references text not animated with Fly In from left: {unmatched}',
partial
```

## A.7. Rubric Correction by Cross-Evaluation

Although LLMs provided the initial rubric that was refined by human annotators, we further paired the annotators to cross-evaluate each other's prepared rubrics and make targeted corrections. These edits included revising the rubric functions or VLM calls, adding or removing nodes from the rubric tree, and rewording task goals for clarity. In addition, we experimented with different scoring strategies to better capture the trade-offs between critical and non-critical node scoring, ultimately selecting the approach that best balanced fairness and robustness.

---

**Cross-Evaluation Instructions**

The goal of this round of reviews is to cross-check tasks and rubrics and address any remaining minor issues with these.

We have two strategies for scoring that we are testing out:

1. **Mind2web 2**: This is the scoring strategy introduced in the mind2web 2 paper which gates on critical node success and averages non-critical node scores.

2. **Default**: This is a different scoring strategy where we take a weighted average between critical and non-critical node scores when calculating parent scores.

Please try out various levels of task completion progress (no progress, various styles of partial progress, task complete) with both scoring strategies and vote on which method wins for this task/rubric, or if there is a tie.

---

# B. Qualitative Meta-Evaluation of Rubric Performance

To assess the reliability and validity of our automated rubric scoring system, we conducted a qualitative meta-evaluation comparing human expert judgments with automated rubric scores across a diverse set of task completion examples. This analysis revealed several categories of agreement and disagreement patterns that provide insight into the strengths and limitations of our evaluation framework.

---

### Meta-Evaluation Instructions

The goal of this round is to evaluate the task rubrics from the polished task rubrics from the previous round, under various levels of task completion and see if this correlates with what you expect. Please follow the instructions below. For every task, create modified decks/test cases for the following levels of task completion:

1. **Category 1 (expected score: 0)**: No progress is made towards task completion—this can be the original deck or completely irrelevant changes.

2. **Category 2 (expected score: $> 0$ and $< 0.5$)**: Partial completion but closer to a score 0 than to 1.

3. **Category 3 (expected score: $\geq 0.5$ and $< 1$)**: Partial completion but closer to a score of 1 than to 0.

4. **Category 4 (expected score: 1)**: Perfect task completion.

Note, for some tasks, it may not make sense to create both Category 2 and Category 3 test cases (e.g., if you expect scores to only be 0, 0.5, and 1 based on the degree of task completion, it may not make sense to include a Category 2 test case). In such cases, you can omit one of the categories. Even rarer are tasks where it would not make sense to provide any partial credit. In this case, you can omit both Category 2 and Category 3. Please omit categories sparingly.

*Tip:* Before grading, you may want to save the original PPTX file locally and grade with this locally saved file—to avoid errors where you may see many files being modified when you have not made any changes at all.

---

### B.1. Categories of Human-Rubric Agreement and Disagreement

We identified four primary categories of disagreement between human evaluators and the automated rubric scoring system:

1. **Human Error**: Cases where human evaluators misunderstood task requirements or made inadvertent errors during assessment (no example collected due to self-evident nature).

2. **Subjective Task Elements**: Disagreements stemming from legitimate differences of opinion on subjective aspects such as color suitability, positioning preferences, or aesthetic choices.

3. **LLM Hallucination**: Instances where the non-deterministic language model component of the rubric scorer generated inaccurate assessments or identified non-existent elements.

4. **Partial Completion Granularity**: Cases where both human and automated evaluators agreed that partial progress was made, but disagreed on the specific degree of completion (e.g., distinguishing between "Some Progress" and "Significant Progress").

Our evaluation framework employs four completion categories: No Progress (0.0), Some Progress (0.33), Significant Progress (0.67), and Perfect Completion (1.0).

### B.2. Examples of Human-Rubric Agreement

B.2.1. PERFECT AGREEMENT CASE (FIGURE 9)

**Task**: Insert a small table on slide 3 showing BMI ranges categorized by male and female and for ages (2-5, 6-11, 12-19, 20+) using $<$, $>$, and $-$ to denote ranges on slide 3.

**Human Assessment**: Perfect Completion
**Rubric Score**: Perfect Completion (1.0)

**Evaluation Reasoning**: The criterion received a perfect score because all requirements for inserting a BMI table on slide 3 were successfully met. A properly formatted table containing male and female BMI ranges with the correct formatting symbols was found on the designated slide. The table was appropriately sized and positioned without being obtrusive to the existing content, and no unnecessary changes were made to other slides in the presentation.

### B.2.2. SIGNIFICANT PROGRESS AGREEMENT CASE (FIGURE 10)

**Task**: Add a text box in Slide 4 explaining what 'prostomium' means and position it above the word 'prostomium' on the diagram.

**Human Assessment**: Significant Progress
**Rubric Score**: Significant Progress (0.67)

**Evaluation Reasoning**: The criterion received a score of 0.67 because while the agent successfully added a text box with a proper explanation of "prostomium" to Slide 4, it failed to position the text box correctly. The text box was placed in the lower center portion of the slide instead of above the word "prostomium" as it appears in the earthworm diagram on the left side. Since positioning is a critical requirement and the agent missed this key aspect, the overall performance was significantly impacted despite getting the content and presence of the text box right.

### B.3. Examples of Human-Rubric Disagreement

### B.3.1. VLM HALLUCINATION AND SUBJECTIVE DISAGREEMENT (FIGURE 11)

**Task**: Add appropriate emojis icons next to each social class type on slide 8.

**Human Assessment**: Perfect Completion
**Rubric Score**: Significant Progress (0.5)

**Sources of Disagreement**: This case exemplifies the intersection of LLM hallucination and subjective assessment differences. The automated rubric reasoning stated: "The criterion received a score of 0.50 primarily due to significant issues with emoji positioning and appropriateness. While emojis were successfully added to slide 8 without affecting other slides, the visual analysis revealed that the emojis weren't properly positioned next to the social class types as required. Additionally, some emoji choices were problematic - most notably using a king emoji to represent a 'classless society,' which is contradictory since royalty represents the opposite of a classless system."

This is a good example of two different categories of failure modes – subjective differences (the human believed that the positioning of the emojis was correct, the model disagreed) and model hallucinations (the model misinterpreted a farmer emoji as a king emoji).

### B.3.2. PARTIAL COMPLETION GRANULARITY DISAGREEMENT (FIGURE 12)

**Task**: Replace images on slide 5 with appropriate text placeholders.

**Human Assessment**: Significant Progress
**Rubric Score**: Some Progress (0.45)

**Source of Disagreement**: Both the human evaluator and the rubric agree that the task was partially completed, but disagree on the degree of completion. The automated rubric reasoning noted: "The criterion received a low score because while all images were successfully deleted from slide 5, only half of the required text content was added to the replacement textbox. The textbox contained 'Image Placeholder - Eye Examination 1' but was missing 'Image Placeholder - Eye Examination 2.' Additionally, unnecessary changes were made to the slide beyond what was required, with multiple extra shapes being added when only one textbox replacement was needed."

The human evaluator weighted the successful image deletion and partial text replacement more favorably, viewing the completion as crossing the threshold from "Some Progress" to "Significant Progress."

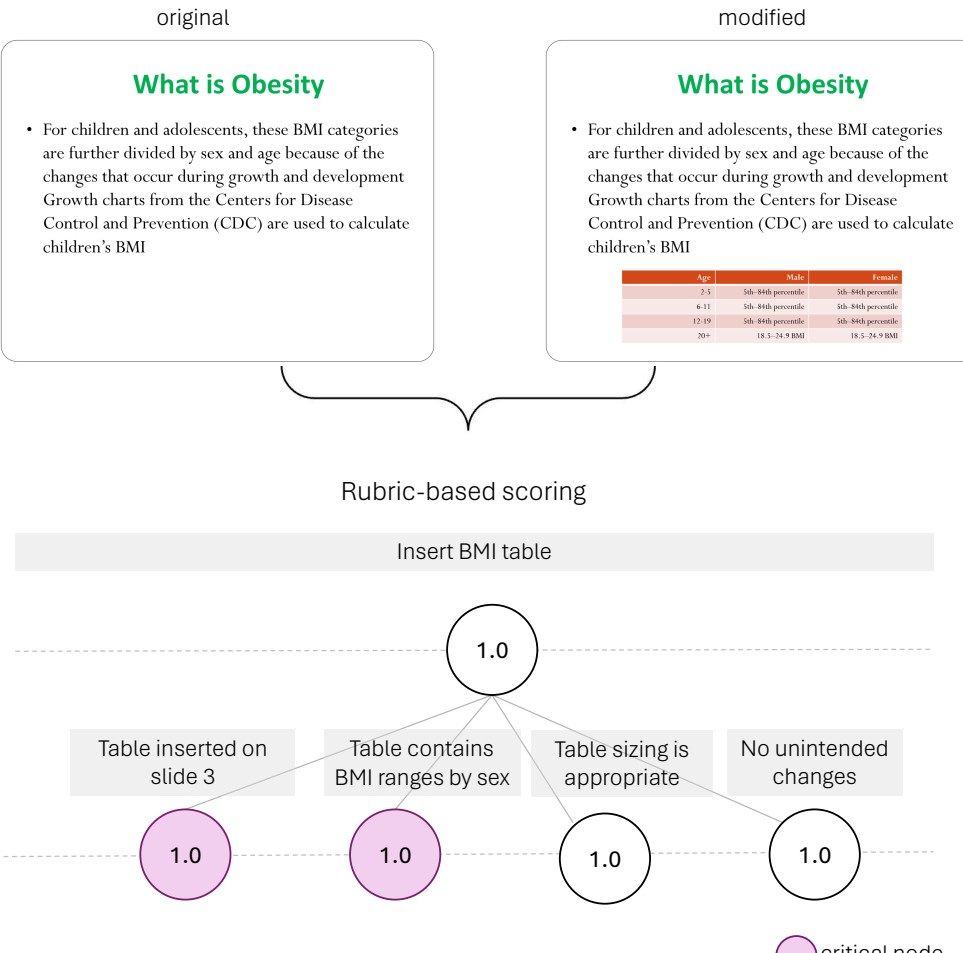

*Figure 9.* Example of perfect human-rubric agreement: BMI table insertion task completion

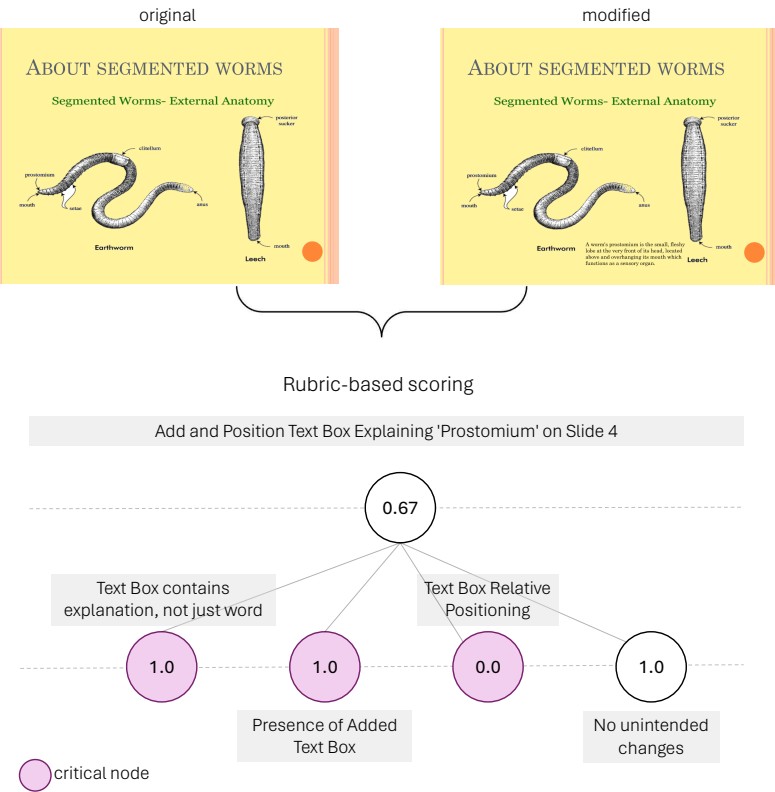

*Figure 10.* Example of human-rubric agreement on significant progress: Prostomium text box task

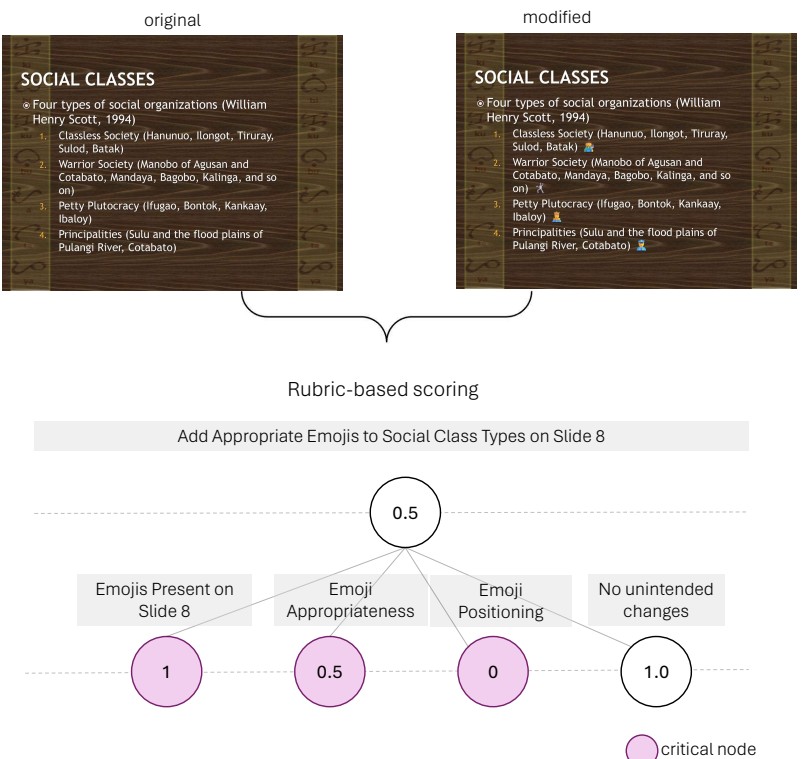

*Figure 11.* Example of VLM hallucination and subjective disagreement: emoji placement task

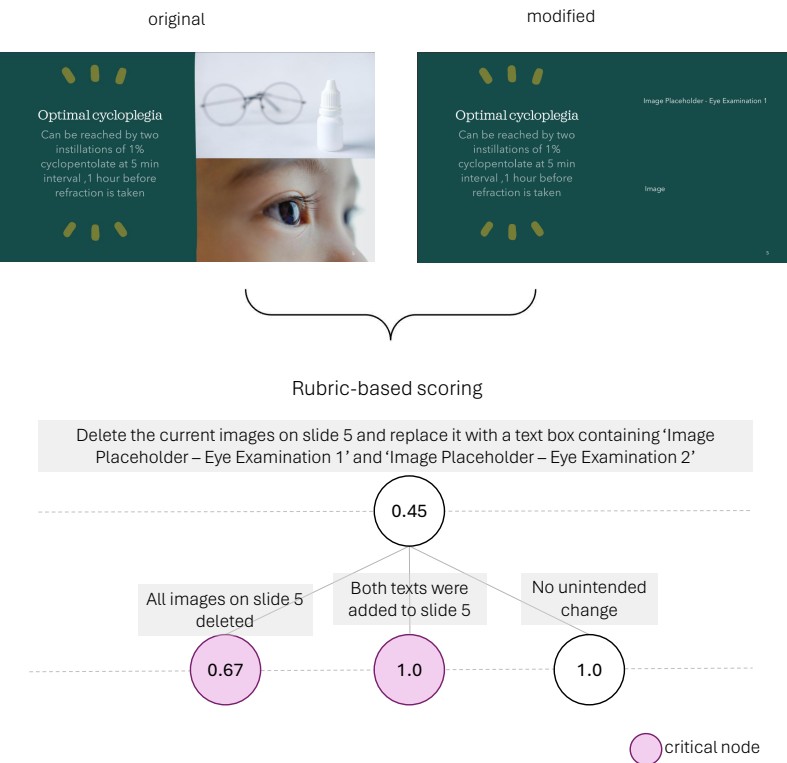

*Figure 12.* Example of partial completion granularity disagreement: Image replacement task

## B.4. Implications for Rubric Reliability

These meta-evaluation examples demonstrate that while our automated rubric system achieves reasonable alignment with human judgment, there are still sources of disagreement that persist. Some of these are a consequence of LLMs' subjectiveness. But subjective tasks are unavoidable in productivity applications and (1) Including them paints a more complete picture of how well agents do on these tasks; (2) Even human judges can disagree on how they evaluate subjective aspects of a task.

## B.5. Rubric Variance Analysis

Since we employ VLM calls for certain visual or subjective checks in our rubrics, we also perform a variance analysis, where for each non-deterministic task we rerun evaluation five time for the same set of rollouts for both the CLAUDE-4-SONNET and COMPUTER-USE-PREVIEW models. Table 7 shows the variance metrics for these runs, showing very low variance and good stability across runs.

By default we used CLAUDE-4-SONNET for VLM calls in the rubrics. To understand variance of rubrics judgements when using different models as VLMs, we reran evaluation of the CLAUDE-4-SONNET trajectories with GPT-4.1 on tasks requiring VLM judgements, and found minimal deviation. In particular, the Mean Absolute Error (MAE) between the judgements from the two models was 0.1 with 78.3% of the scores lying within ±0.1 of each other.

*Table 7.* Comparison of shared stability metrics across 5 evaluation runs on the 61 PPTArena tasks with non-deterministic rubrics.

| Model | Mean Var | Median Var | Mean CV |
|---|---|---|---|
| Computer-Use-Preview | 0.0008 | 0.0000 | 0.073 |
| Claude-4-Sonnet | 0.0062 | 0.0000 | 0.093 |

## C. PPT-EVAL Feature Summary

Fig. 8 summarizes the main differences between PPT-EVAL and other related benchmarks.

*Table 8.* Comparison of related benchmarks on features central to PPT-EVAL.

| Benchmark | PowerPoint | GUI Interaction | Partial credit | Difficulty tiers | NL feedback |
|---|---|---|---|---|---|
| MIND2WEB 2 (Gou et al., 2025) | ✗ | ✓ | ✓ | ✗ | ✗ |
| OSWORLD (Xie et al., 2024) | ✗[*] | ✓ | ✗ | ✗ | ✗ |
| WINDOWSAGENTARENA (Bonatti et al., 2025) | ✗ | ✓ | ✗ | ✗ | ✗ |
| OFFICEBENCH (Wang et al., 2024) | ✗ | ✗ | ✗ | ✗ | ✗ |
| SHEETAGENT–SHEETRM (Chen et al., 2024) | ✗ | ✗ | ✓ | ✗ | ✗ |
| PPTC (Guo et al., 2024) | ✗ | ✗ | ✗ | ✗ | ✗ |
| SLIDESBENCH (Ge et al., 2025) | ✗ | ✗ | ✗ | ✗ | ✗ |
| **PPT-EVAL** | ✓ | ✓ | ✓ | ✓ | ✓ |

[*] Uses LibreOffice Impress for presentation-related tasks.

## D. Rubric Inference Prompts

We use the following prompt to bubble-up natural language explanations from child nodes to parent nodes and create an overall explanation for a rubric score.

---

**Rubric Aggregation Prompt**

You are evaluating a rubric criterion called '{self.name}': {self.description}
This criterion has the following sub-criteria with their scores and reasons:

```
{% for child_info in children_info %}
- {{ child_info.name }} ({{ child_info.label }})
    Score: {{ child_info.score }}
    Description: {{ child_info.description }}
    Reason: {{ child_info.reason }}
{% endfor %}
```

The overall score for '{self.name}' is {self.score:.2f}.
Rubric scoring rules:

- If both critical and non-critical children exist:

$$\text{overall} = \max(0, \text{average(critical)} - \lambda \times (1 - \text{average(non-critical)})),$$

```
{% if self._last_non_critical_weight %}
with lambda = {{ self._last_non_critical_weight }}
{% endif %}
```

- Otherwise (all children critical or all non-critical): average of all children

Please provide a concise reason (1-5 sentences) explaining why this criterion received a score of {self.score:.2f}, referencing the relevant sub-criteria and their performance. Focus on the most important factors that determined the score. Make the the reason more natural language and human-like rather than formulaic, and avoid including numerical scores in the reasoning.

---

# E. Additional Benchmark Run Analyses

## E.1. Success and Failure Clusters

To further characterize both failures and successes across the GUI agent models, we performed an additional clustering analysis, shown in Fig. 13. The top heatmap groups tasks into high-level intent categories and shows model success rates within each category. The bottom heatmap clusters the natural-language rubric explanations for failed tasks into common failure patterns and shows the proportion of each model's failures associated with each pattern.

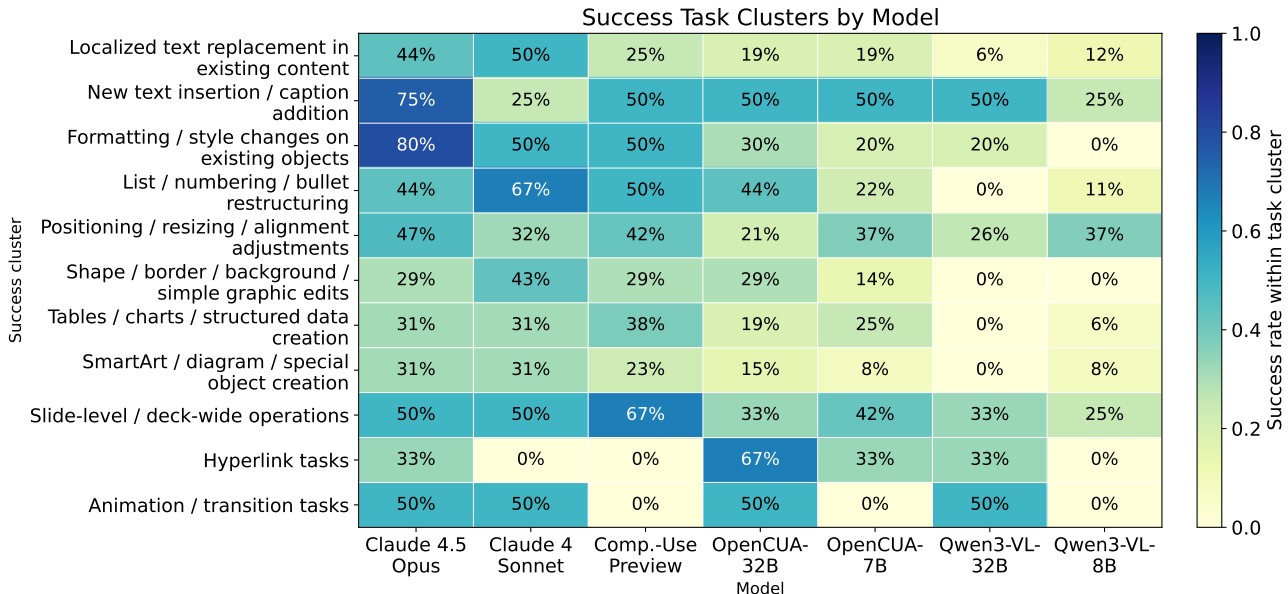

*(a)* Success rate within each task cluster, by model.

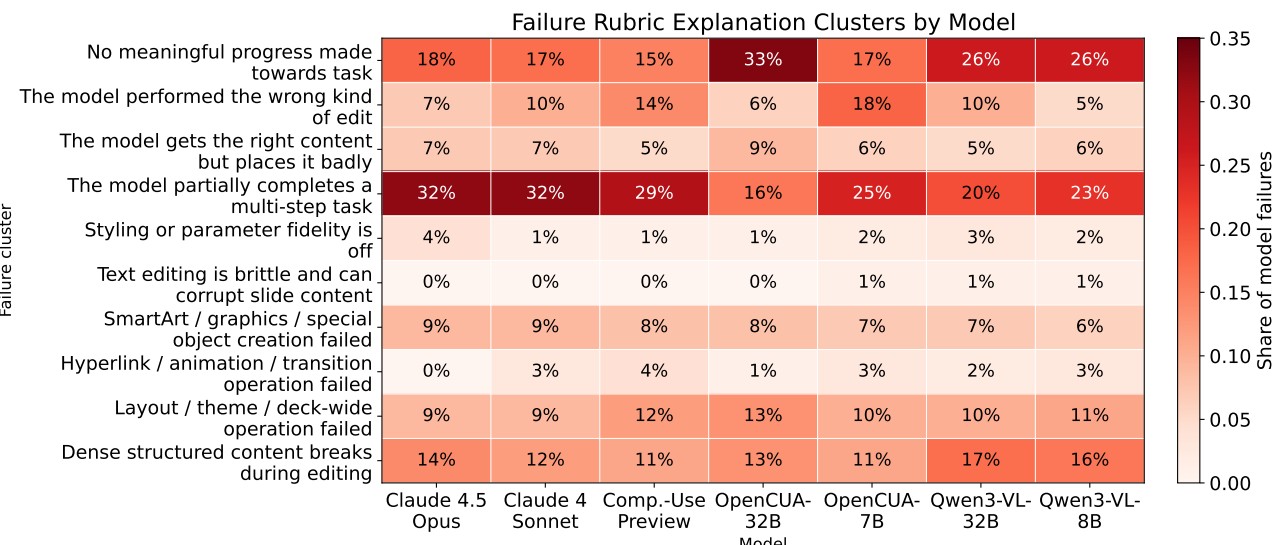

*(b)* Share of model failures attributed to each failure reason cluster.

*Figure 13.* Per-cluster success and failure breakdown across models.

## E.2. Variance Across Runs

To better understand performance variability across repeated evaluations, we conducted multiple benchmark runs for the human baseline, API baseline, and closed-source models. Table 9 reports the sample standard deviation of Success Rate, Average Score, and Average Steps across runs. The corresponding mean performance values are reported in Table 1.

*Table 9.* Cross-run variance on PPT-Eval. Each cell reports the sample standard deviation of *Success Rate (SR) / Avg. Score / Avg. Steps* across repeated benchmark runs. Human baseline, API baseline, and closed-source models were evaluated multiple times. Corresponding mean performance values are reported in Table 1. Human baseline results are computed from five runs, while API-based and closed-source model results are computed from three runs. Corresponding mean performance values are reported in Table 1.

| Model | Overall (120) | Easy (51) | Medium (39) | Hard (30) |
|---|---|---|---|---|
| Human Baseline | 0.04 / 0.03 / – | 0.07 / 0.06 / – | 0.03 / 0.03 / – | 0.05 / 0.06 / – |
| API Baseline (Claude-4.5-Opus) | 0.03 / 0.01 / – | 0.02 / 0.01 / – | 0.04 / 0.01 / – | 0.07 / 0.03 / – |
| *Closed Models.* | | | | |
| Claude−4.5−Opus | 0.02 / 0.02 / 1.26 | 0.03 / 0.03 / 1.27 | 0.03 / 0.04 / 2.06 | 0.07 / 0.01 / 1.23 |
| Claude−4−Sonnet | 0.02 / 0.07 / 2.52 | 0.04 / 0.04 / 1.52 | 0.08 / 0.16 / 1.92 | 0.03 / 0.13 / 5.07 |
| Computer−Use−Preview | 0.04 / 0.02 / 1.59 | 0.04 / 0.03 / 2.01 | 0.05 / 0.02 / 1.52 | 0.05 / 0.03 / 1.19 |

## E.3. Running Cost & Time

With concurrency set to 3, we find that a benchmark run takes ∼3-4 hours to run. A single GUI agent run with frontier API-access models takes upward of $50, with CLAUDE-4.5-OPUS costing $54 and COMPUTER-USE-PREVIEW costing $62.

## F. Agent Trajectory Examples

In this section, we show some examples of agent trajectories as they perform tasks on the sandbox environment.

### F.1. Computer-Use-Preview

Fig. 14 shows how OpenAI's computer-use-preview (CUA) model was able to successfully solve a task of "Hard" difficulty. The goal of the task was to "Crop the architectural diagram image on slide 8 to show only the top two rows of buildings", and the model was able to do just that in under 10 steps. The evaluator scored the output as 1.0 (Perfect Completion).

On the other hand, we also demonstrate a case where the model fails at a similarly "Hard" task - "Adding slide numbers to every slide except the title slide." While the model does turn on the footer, it does not select the slide number option, receiving a score of 0.5. This is shown in Fig. 15.

### F.2. Claude-4-Sonnet

On the same task "Add slide numbers to all slides except the title slide" that CUA fails, Claude successfully executes with a perfect score of 1.0, as shown in Fig. 16. Since Claude uses a different action space, we only save the screenshots and not the computer actions in the trajectory.

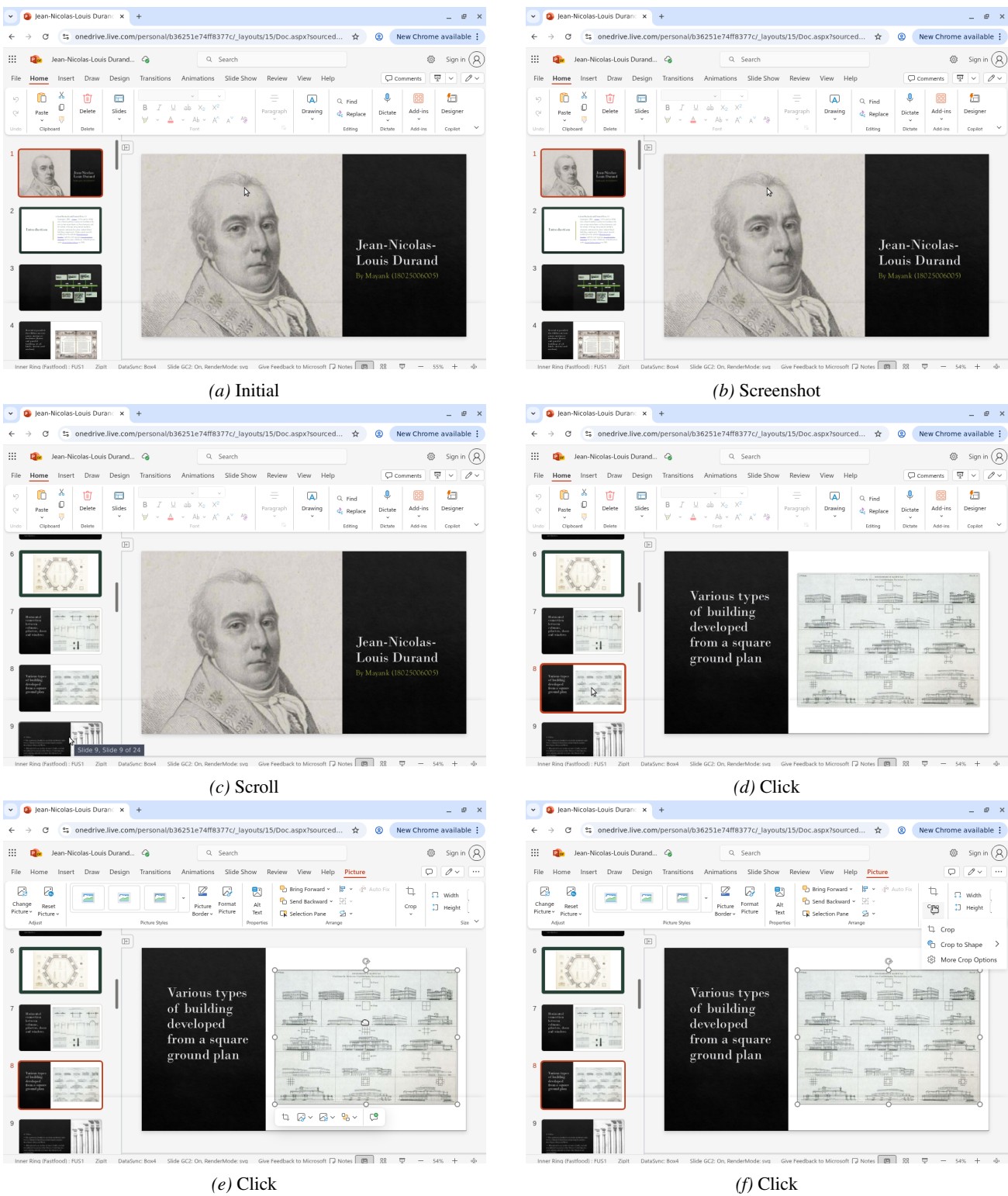

*Figure 14.* CUA's successful trajectory for the task 'Crop the architectural diagram image on slide 8 to show only the top two rows of buildings'.

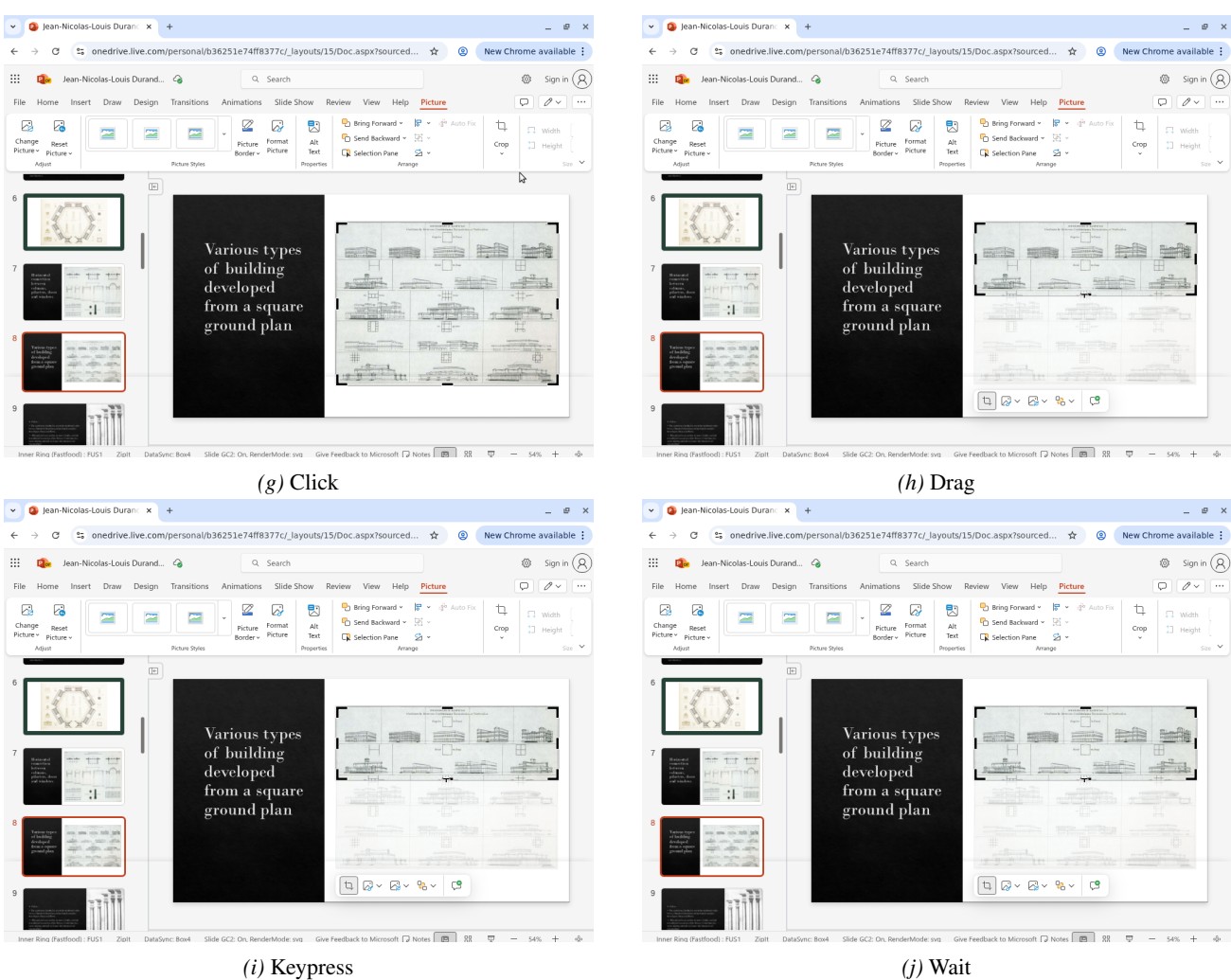

*(g)* Click

*(h)* Drag

*(i)* Keypress

*(j)* Wait

*Figure 14.* CUA's successful trajectory for the task 'Crop the architectural diagram image on slide 8 to show only the top two rows of buildings' (continued).

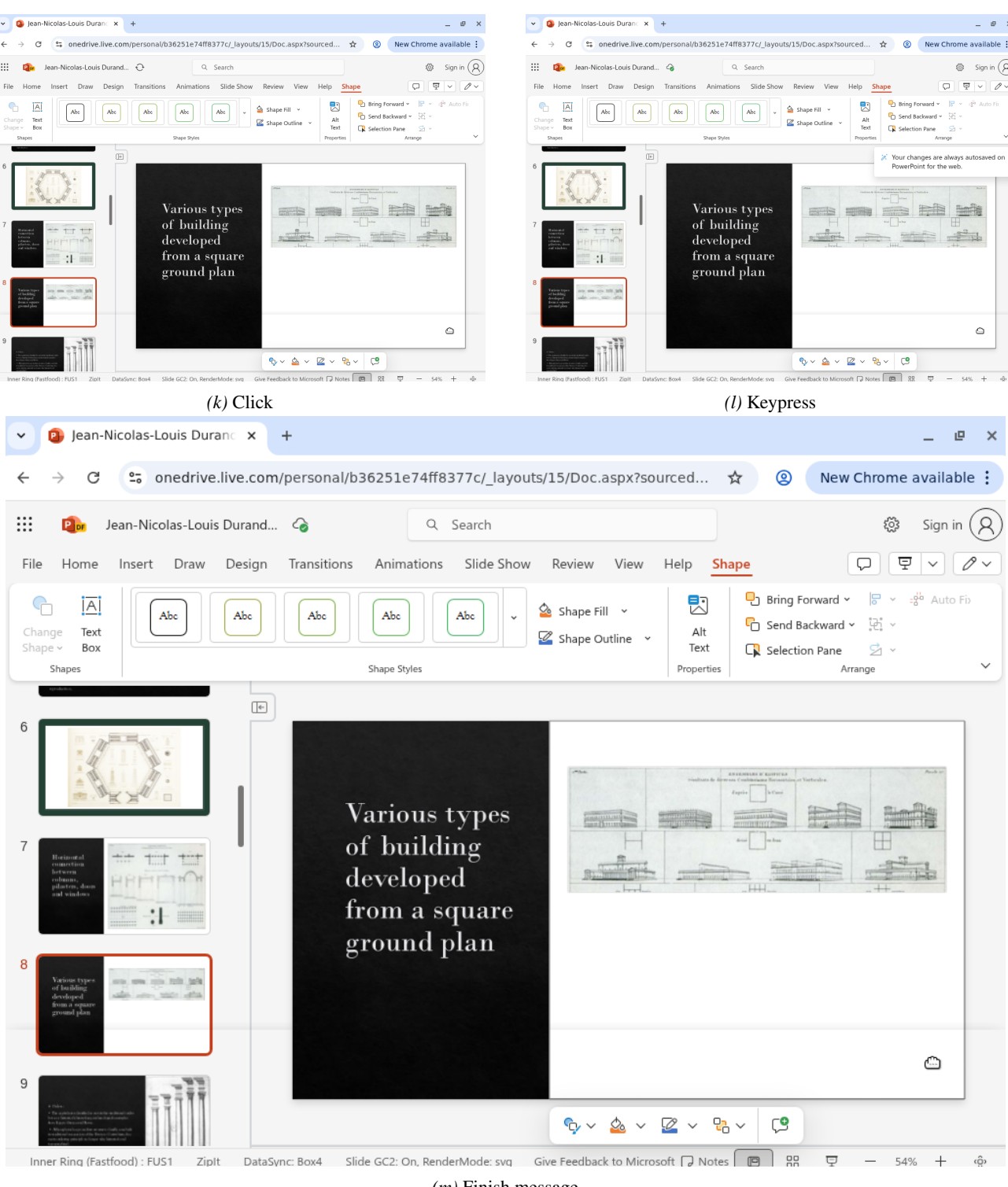

*(k)* Click

*(l)* Keypress

*(m)* Finish message

*Figure 14.* CUA's successful trajectory for the task 'Crop the architectural diagram image on slide 8 to show only the top two rows of buildings' (continued).

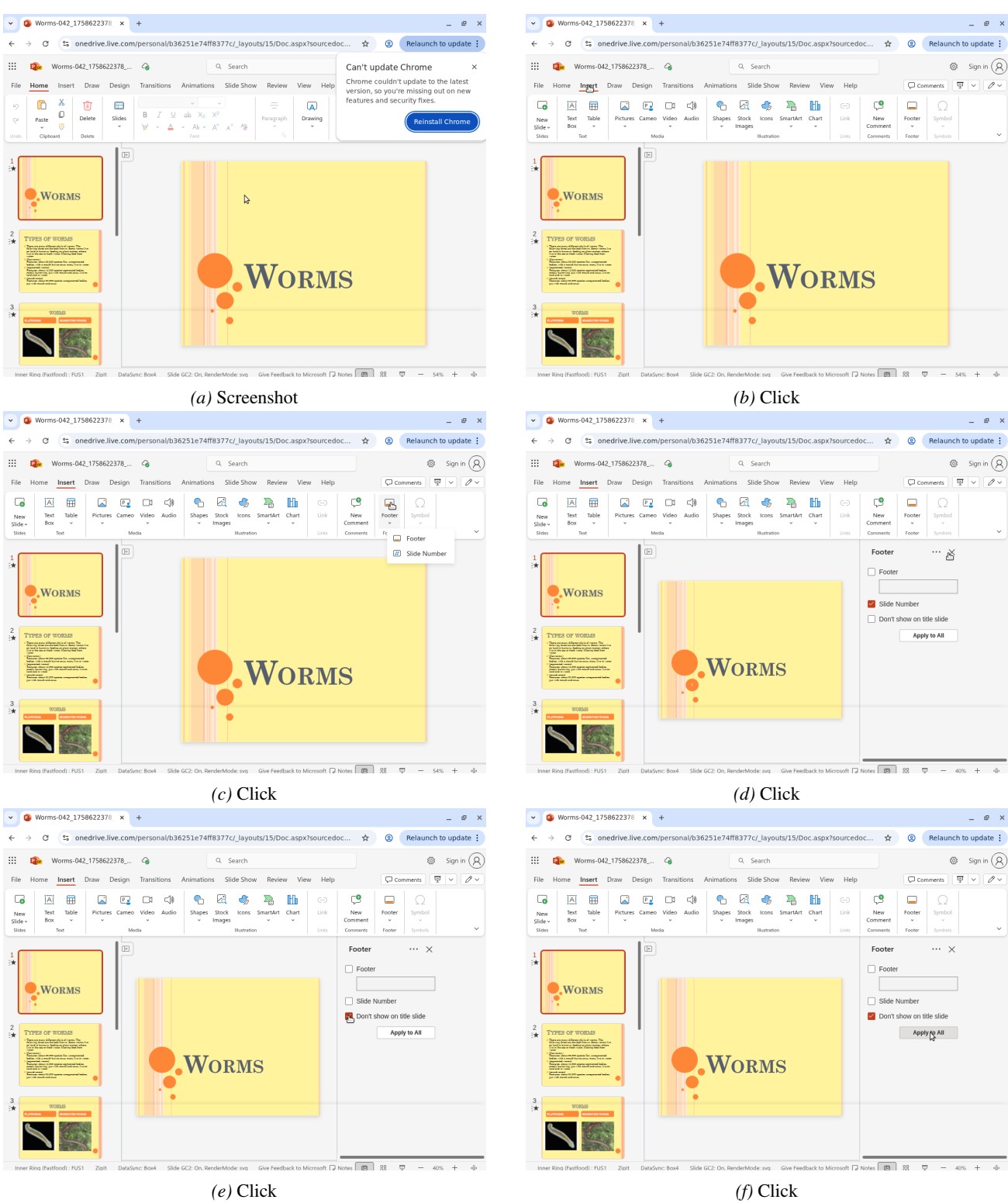

*Figure 15.* CUA's failed trajectory for the task 'Add slide numbers to every slide except the title slide'.

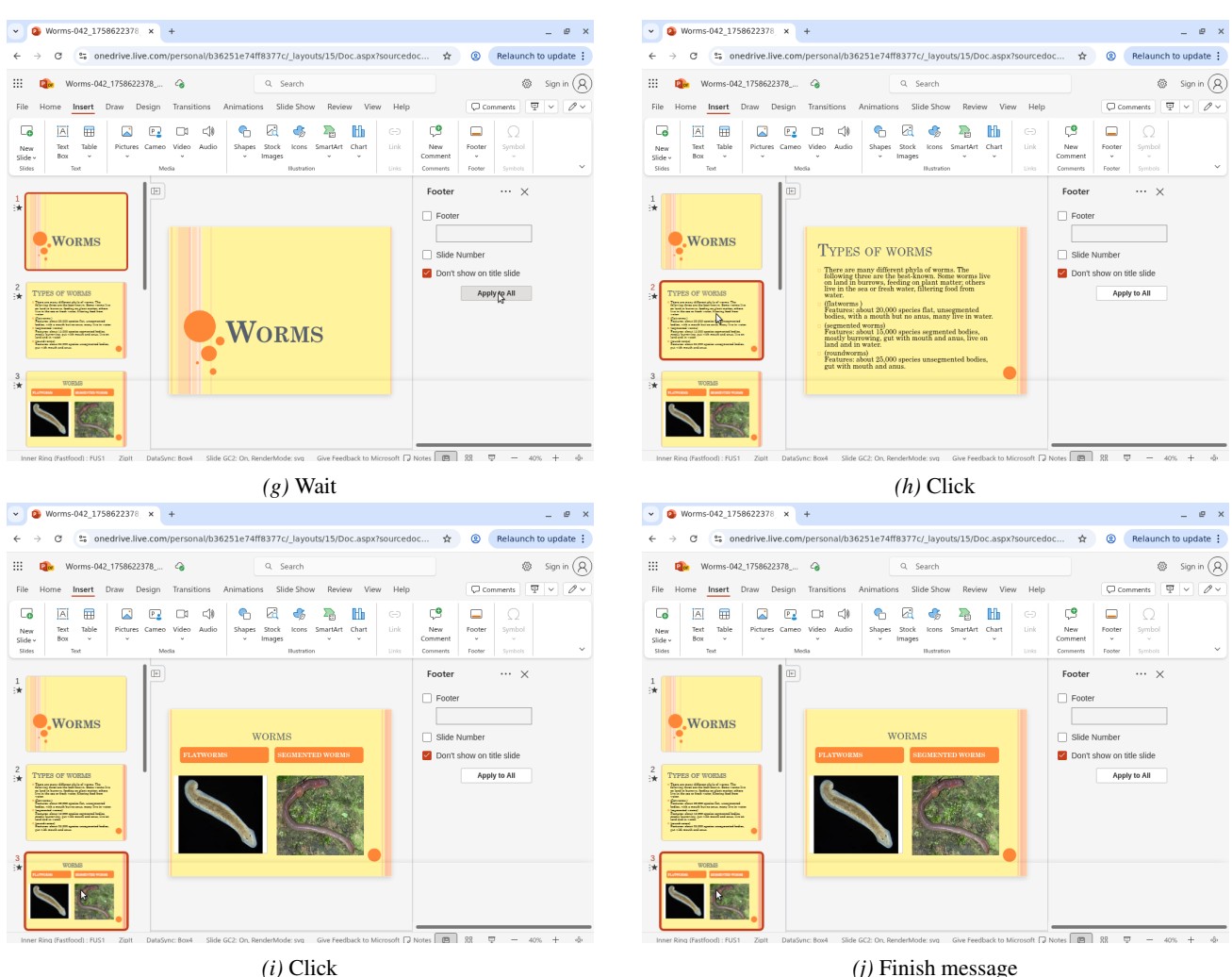

*(g)* Wait

*(h)* Click

*(i)* Click

*(j)* Finish message

*Figure 15.* CUA's failed trajectory for the task 'Add slide numbers to every slide except the title slide' (continued).

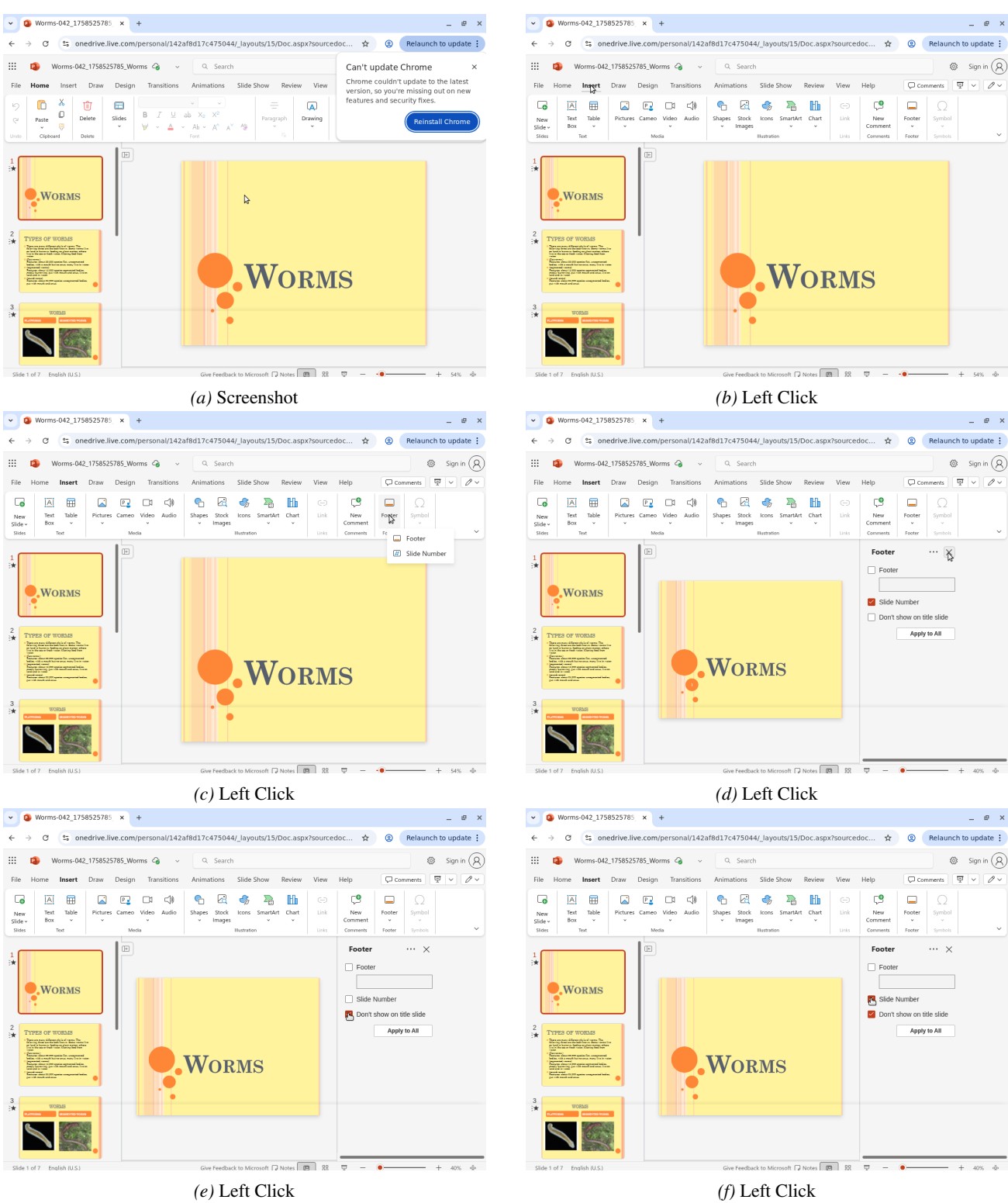

*(a)* Screenshot

*(b)* Left Click

*(c)* Left Click

*(d)* Left Click

*(e)* Left Click

*(f)* Left Click

*Figure 16.* Claude's successful trajectory for the task "Add slide numbers to all slides except the title slide."

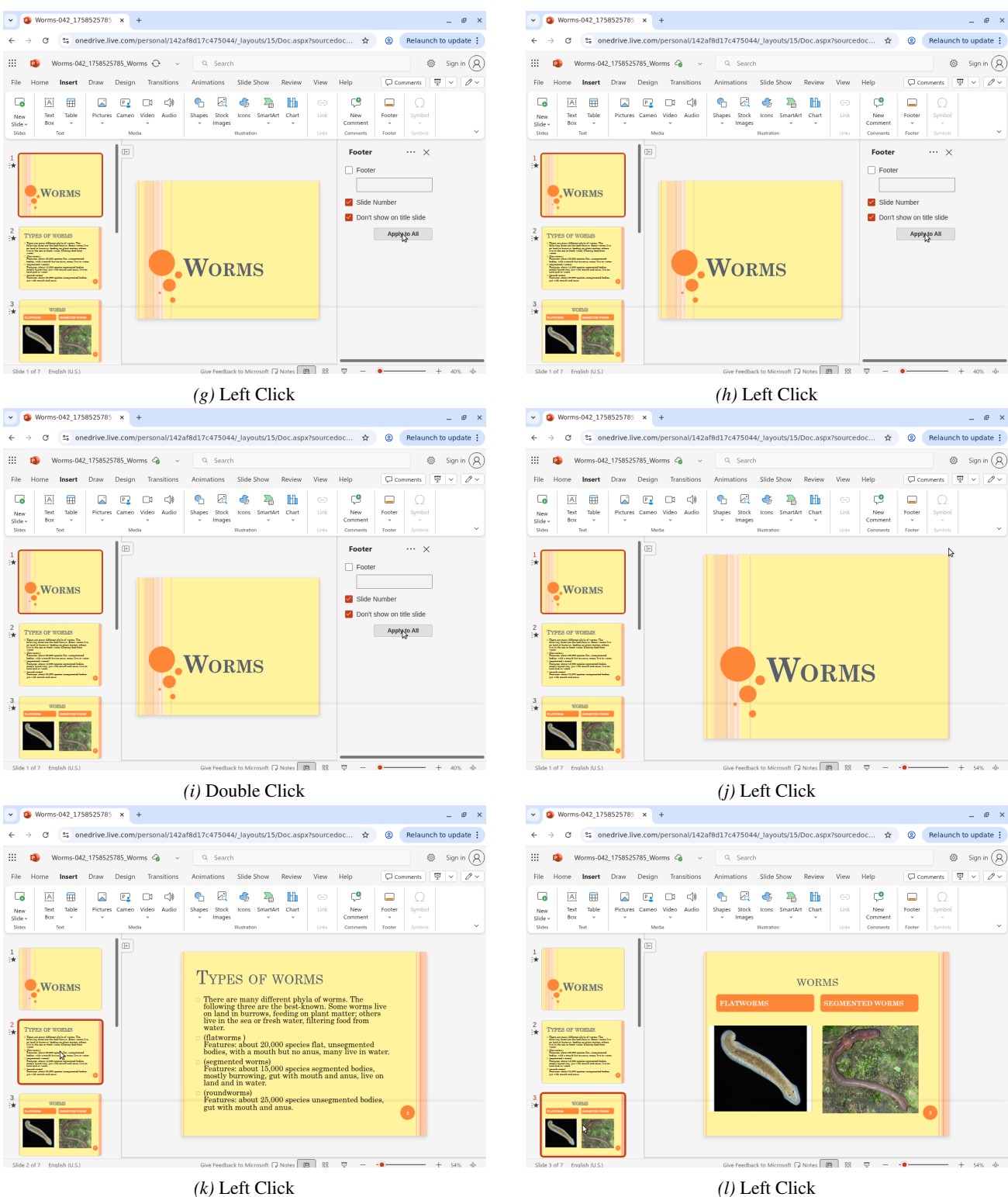

*Figure 16.* Claude's successful trajectory for the task "Add slide numbers to all slides except the title slide" (continued).

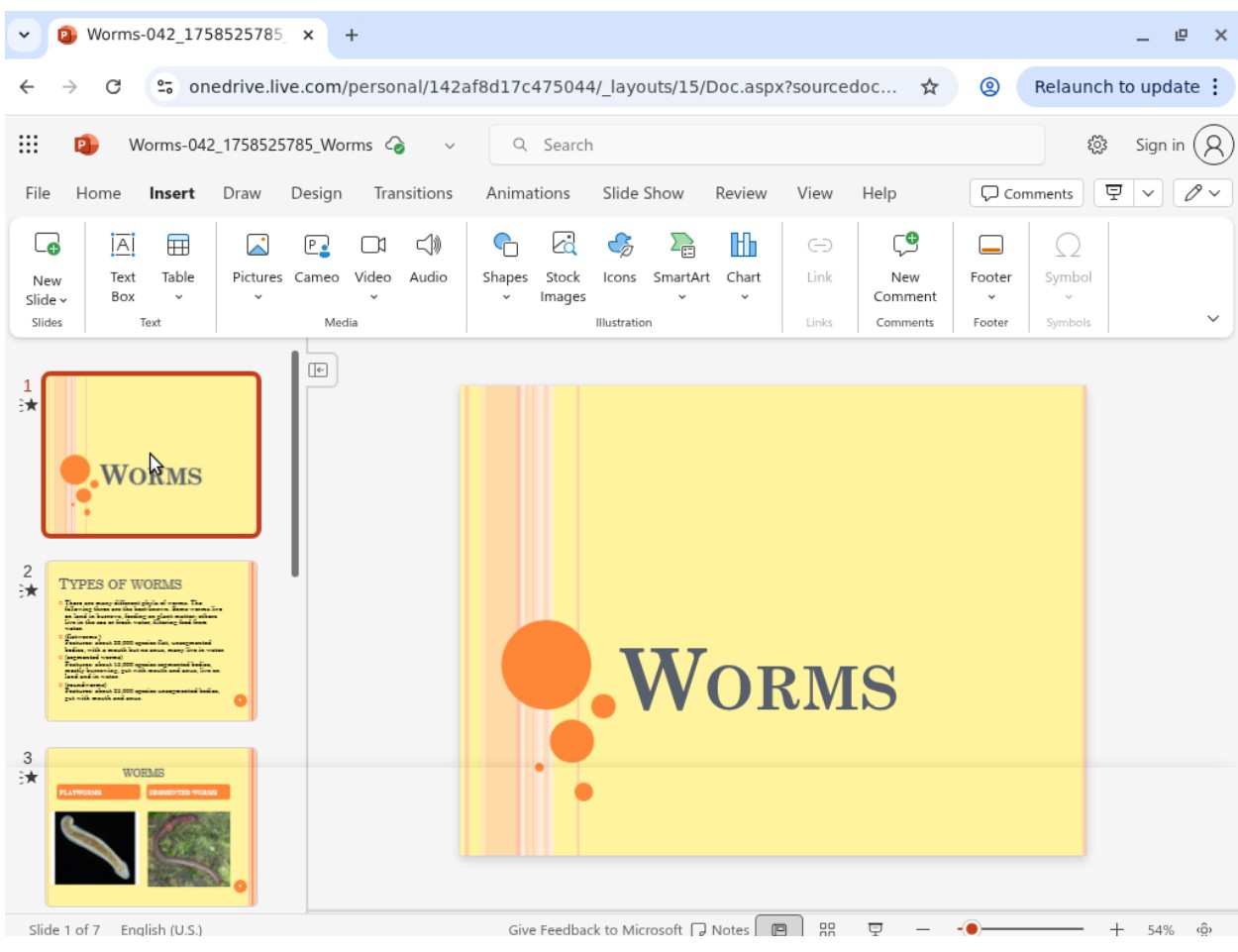

*(m)* Finish

*Figure 16.* Claude success steps (continued).

# G. API Baseline Implementation

To provide a comparable evaluation using an API based approached, we added support for CLI-based models that work by manipulating a copy of the input file offline. We set up a code path that allows CLI based agents like Claude Code to attempt these tasks.

## G.1. Setup

To provide a fair comparison, we made use of Claude Code, a popular CLI based harness for Anthropic models. The execution runs used Claude-4-5-Opus and utilized the official `pptx` skill from the Anthropic repository. The CLI agent is provided with a single task workspace – a folder containing the task to be executed, the instructions for output format, the input files and a folder for the output file. Multiple agents can be triggered across multiple tasks at the same time for concurrent benchmark execution.

## G.2. System Prompt

The system prompt used by the agent is also provided below:

### Task Proposal Prompt

```
# CLAUDE.md      PPTEval per-task workspace

You are running inside a **single-task workspace** for the PPTEval
benchmark. Each invocation handles exactly one task. The harness has prepared
this directory with everything you need.

## Layout

```
.
        TASK.md                    # the task goal (read this first)
        OUTPUT_INSTRUCTIONS.md     # exact input/output file paths
        inputs/<file>.pptx         # the source presentation (read-only)
        output/                    # write the modified file here
        .claude/skills/pptx/       # OOXML / python-pptx skill
```

## How to work

1. **Read `TASK.md` and `OUTPUT_INSTRUCTIONS.md` first.** They define the goal
   and the exact path you must write your output to. Follow the output path
   literally    the grader looks for that path and nothing else.
2. **Treat `inputs/` as read-only.** Copy the file before modifying it.
3. **Write your result to `output/<task_id>.<ext>`** exactly as specified.
   If the file is missing or named differently, the task scores 0.
4. **When the output file exists and reflects the requested changes, exit.**
   Do not leave background processes running.

## Tools you have

- The `pptx` skill in `.claude/skills/pptx/` (SKILL.md, plus OOXML
  pack/unpack scripts and python-pptx workflows). Read `SKILL.md` for
  guidance on:
  - text extraction (`python -m markitdown ...`)
  - unpacking/repacking pptx files for raw XML edits
  - common edit patterns (colors, fonts, text replacement, slide
    rearrangement, thumbnails)
- `python` with `python-pptx` and `markitdown` already installed.
- Standard shell utilities.

## Task interpretation tips
```

```
-  On slide N       slide N, **1-indexed** as it appears in PowerPoint.
-  Change the title      the title placeholder of the slide.
-  Change color to X       apply to fill / font color as the wording implies.
-  Add a text box / bullet / shape      create a new element; do not replace
   existing content unless asked.
- Make **only** the changes requested. Do not restructure unrelated slides
  or add commentary.

## Scoring

The grader compares your `output/<task_id>.<ext>` against the original using
a rubric defined per task. Strict success requires `score == 1.0`, so make
sure your edits match the task description precisely.

## Verify-then-retry loop (REQUIRED)

After you produce `output/<task_id>.<ext>`, **do not exit immediately**.
Run a verification pass:

1. **Re-read `TASK.md`** and list every concrete requirement (slide N,
   target object, attribute, exact value, etc.).
2. **Open your output file** with `python-pptx` (or unpack the XML if the
   change is style/layout) and programmatically confirm each requirement
   is satisfied:
   - For text changes: read the run text on the targeted shape and check
     it matches exactly (including capitalisation and punctuation).
   - For color/font/size changes: read the relevant property and assert
     it equals the requested value.
   - For shape/image add/remove/resize: confirm the shape count, type,
     position, or dimensions.
   - For slide reorder/duplicate/delete: confirm slide count and order.
3. **Diff against `inputs/<file>.pptx`** to make sure you did **not**
   change anything outside the task scope.
4. If any check fails, **fix the file and re-verify**. Allow yourself up
   to 3 verification retries before giving up.
5. Only exit once all verification checks pass, or after 3 retries     log
   what failed so the run is debuggable.

Keep the verification script in `sandbox/` or stdout; you dont need to
ship it. The harness only reads `output/<task_id>.<ext>`.

## Common failure modes to guard against

- Writing the output under the wrong filename or extension.
- Modifying `inputs/` in place instead of copying.
- Editing the wrong slide because of 0-indexed vs 1-indexed confusion.
- Replacing a placeholders entire content when the task only asked to
  add/append.
- Forgetting to save (`prs.save(...)`).
- Leaving a `.pptx` open via `python-pptx` so the file is truncated.
- Skipping the verification step and exiting on the first save.
```

