# OpenReview forum: "PPT-Eval: A Benchmark for Computer-Use Agents on PowerPoint Tasks"
_ICML.cc/2026/Conference — ICML 2026 regular_

### Official Review · Reviewer_cQZG · 2026-03-01

**Soundness:** 3
**Presentation:** 3
**Significance:** 3
**Originality:** 3
**Overall Recommendation:** 4
**Confidence:** 4

**Summary:**

The authors introduce PPT-EVAL, a GUI-based benchmark for evaluating computer-use agents on realistic PowerPoint editing tasks. The benchmark contains 120 tasks across 12 real-world decks and evaluates agents through full GUI interaction rather than restricted APIs. The core contribution is a tree-structured rubric system that supports partial credit and penalizes unintended edits.

PowerPoint editing is multimodal, sequential, and structurally constrained, making it a strong stress test for real-world GUI agents. Binary pass/fail metrics are insufficient in this setting, and the rubric-based evaluation is designed to capture meaningful intermediate progress. The paper shows that even frontier models remain far from human-level performance.

The best models achieve ~43% success rate, compared to 81% for humans. Agents struggle especially with animations, layouts, and complex multi-element edits. The results highlight that GUI autonomy and spatial reasoning are still brittle.

**Compliance With Llm Reviewing Policy:**

Affirmed.

**Key Questions For Authors:**

1. How sensitive are rubric scores to the specific LLM/VLM judge used? Have you tested cross-judge consistency?

2. How would the evaluation behave if λ in the aggregation rule changes? Is the choice of 0.3 empirically justified or heuristic?

3. What is the approximate token or monetary cost per trajectory (including rubric evaluation)? Would this benchmark be practical for RL-style training loops?

4. Do you see a path toward automating rubric construction to scale the benchmark 10×?

5. Have you analyzed trajectory-level failure patterns beyond success rate (e.g., common planning collapse types)?

**Limitations:**

The paper includes a brief impact statement but does not sufficiently discuss deployment risks of unreliable GUI agents or the dependence on LLM/VLM-based judging, and it would benefit from a more concrete analysis of potential misuse, failure severity, and evaluation bias.

**Strengths And Weaknesses:**

## Strengths

1. Realistic and well-motivated problem setting.
This is not a toy benchmark — slide editing is a real productivity workflow with rich multimodal constraints. Evaluating through the actual GUI avoids the artificial simplicity of API-based automation. The environment design makes the benchmark meaningful.

2. Rubric-based evaluation is well thought out.
The tree-structured rubric with critical/non-critical separation is a reasonable solution to partial credit in open-ended tasks. The modified aggregation rule is more sensible than hard gating. The reported correlation with human judgments is strong enough to justify the approach.

3. Clear evidence of capability gaps.
The benchmark convincingly shows that current agents are far from robust in real GUI workflows. The breakdown by task type (layouts, animations, tables) reveals structural weaknesses. This makes the benchmark diagnostic rather than just competitive.

## Weaknesses

1. Rubric complexity and scalability.
Rubric construction required significant manual effort (~150 hours), which does not scale well. Reliance on LLM/VLM judging introduces potential nondeterminism and bias. It is unclear how robust this evaluation remains under judge model changes.

2. Limited scope and scale.
120 tasks across 12 decks is solid but still modest in scale. The benchmark focuses on a single application and does not test cross-application workflows. Generalization beyond PowerPoint is not demonstrated.

3. No cost or efficiency analysis.
The paper does not report inference cost per trajectory, despite using large multimodal models and judge calls. For real deployment or RL training, cost is critical. The absence of cost-performance tradeoff analysis is a noticeable gap.

---

> ### Author Rebuttal · Authors · 2026-03-31
>
> Thank you for your thoughtful feedback and encouraging assessment. We are glad you found the benchmark realistic and well-motivated, and viewed the rubric-based evaluation as a meaningful contribution. We address your concerns below.
>
> > Rubric construction required significant manual effort (~150 hours), which does not scale well. Do you see a path toward automating rubric construction to scale the benchmark 10×?
>
> We agree that 150 hours of human effort is substantial, but we view this not simply as a benchmark weakness, but as an honest reflection of current LLM limitations for generating rubrics for open-ended PowerPoint tasks. One contribution of our paper is to document these limitations in detail, including common failure modes of automated methods and the human effort currently needed to ensure quality (Appendix A.3 and A.5). We believe this analysis is valuable for future work on automated benchmark creation. We will also open-source the task and rubric generation pipeline, which should make extension substantially easier. As LLMs improve, we expect the human effort needed to scale such benchmarks to decrease significantly.
>
> > Reliance on LLM/VLM judging introduces potential nondeterminism and bias. It is unclear how robust this evaluation remains under judge model changes. How sensitive are rubric scores to the specific LLM/VLM judge used? Have you tested cross-judge consistency?
>
> Yes. To measure cross-judge consistency, we reran evaluation for Claude-Sonnet-4 using GPT-4.1 on the 60 tasks requiring LLM/VLM judge calls. The mean absolute error between Claude-Sonnet-4 and GPT-4.1 judgments was 0.1, with 78.3% of scores within ±0.1. We will add these results to the appendix in the camera-ready version.
>
> Appendix B.5 also shows that reevaluating the same rollouts 5 times with the same rubric/judge yields very low variance (mean variance < 0.006).
>
> > Limited scope and scale. 120 tasks across 12 decks is solid but still modest in scale. The benchmark focuses on a single application and does not test cross-application workflows. Generalization beyond PowerPoint is not demonstrated.
>
> While our benchmark contains 120 tasks, this is comparable to established computer-use benchmarks such as AndroidWorld [1] (116 tasks) and Mind2Web 2 [2] (130 tasks). Moreover, even with only 120 tasks, a full benchmark run already takes about 3 hours using three concurrent threads, since computer-use trajectories often take several minutes each.
>
> Cross-application workflows are not our focus, as other benchmarks already cover multi-app settings (e.g., OSWorld, WindowsAgentArena, and OfficeBench; Section 2). Our goal is instead an in-depth benchmark for presentation software, addressing the lack of existing benchmarks for PowerPoint. We will also open-source both the automated pipeline for generating task/rubric drafts and the human refinement instructions to support future extensions.
>
> [1] Rawles, Christopher, et al. "AndroidWorld: A Dynamic Benchmarking Environment for Autonomous Agents." The 13th ICLR.
>
> [2] Gou, Boyu, et al. "Mind2Web 2: Evaluating Agentic Search with Agent-as-a-Judge." NeurIPS Datasets and Benchmarks Track.
>
> > No cost or efficiency analysis. The paper does not report inference cost per trajectory, despite using large multimodal models and judge calls. What is the approximate token or monetary cost per trajectory (including rubric evaluation)?
>
> We agree that the paper would be strengthened by including cost analysis for API-based evaluations. We found that models such as Claude-4-Sonnet cost roughly $50 on average for a full benchmark run. For the camera-ready version, we will report more fine-grained token and monetary costs per model.
>
> > How would the evaluation behave if λ in the aggregation rule changes? Is the choice of 0.3 empirically justified or heuristic?
>
> We set λ = 0.3 due to its intuitive meaning: penalty from performing poorly on non-critical criteria (e.g., making an accidental extraneous change or poor subjective positioning of an added textbox) at most reduces the score by 30% if all critical criteria for the task are satisfied. When developing and revising rubrics, human annotators had the λ value preset to 0.3, and so we did not experiment with different values of λ.
>
> > Have you analyzed trajectory-level failure patterns beyond success rate (e.g., common planning collapse types)?
>
> Thanks for the suggestion. We are happy to add a more detailed analysis of failure reasons per model. One approach is to cluster the rubric-generated natural language explanations into high-level patterns to better characterize model failures and successes. In fact, an earlier draft included such a plot for a subset of models, but we replaced it with Fig. 7 because it provides a more compact summary of failures and successes by task type across all models. We are happy to include the clustering analysis in the appendix for the camera-ready version.

---

> > ### Author Rebuttal · Reviewer_cQZG · 2026-04-05
> >
> > Thank you for the detailed and constructive rebuttal. The additional analysis on cross-judge consistency and variance meaningfully improves confidence in the robustness of the evaluation. However, key concerns around rubric scalability and the sensitivity of the aggregation design (e.g., λ) remain largely unresolved, so my overall assessment is unchanged.

---

> > > ### Author Response · Authors · 2026-04-08
> > >
> > > Thank you for the follow-up. We appreciate the opportunity to provide additional clarifications and results addressing the remaining concerns.
> > >
> > > > Have you analyzed trajectory-level failure patterns beyond success rate (e.g., common planning collapse types)?
> > >
> > > Yes. To better characterize both failures and successes across models, we performed an additional clustering analysis. We provide the resulting figure here: https://anonymous.4open.science/r/ppt-eval-figures-72DE/success_failure_analysis.pdf.
> > >
> > > The left heatmap groups tasks into high-level intent categories and shows model success rates within each category. The right heatmap clusters the natural-language rubric explanations for failed tasks into common failure patterns and shows the proportion of each model’s failures associated with each pattern. We will include this analysis in the appendix together with a discussion. In addition, the paper already provides a breakdown by task-relevant object type in Fig. 7, discussion in Section 5.1, and qualitative trajectory examples in Appendix E.
> > >
> > > > Rubric scalability
> > >
> > > As noted in our original rebuttal, current LLM capabilities are not yet sufficient for fully automatic generation of high-quality rubrics in this domain. Fully LLM-generated benchmark would not meet the bar for evaluation in this domain. We therefore use a semi-automatic rubric construction process to maintain evaluation quality.
> > >
> > > At the same time, we do not view this as a primary weakness of the work. Because this paper introduces a benchmark, the central goal is to provide a rigorous and reliable evaluation suite for PPT tasks, rather than an automatic task or environment generation framework for training. In that setting, human effort devoted to ensuring rubric quality is, in our view, a strength rather than a limitation. This is also supported by our meta-evaluation results in Section 5.2, which show that the rubrics are highly correlated with human judgments.
> > >
> > > We hope these additional results and clarifications help address your concerns, and we would be grateful if you would consider reassessing the score.

---

### Official Review · Reviewer_1vJ7 · 2026-03-11

**Soundness:** 3
**Presentation:** 3
**Significance:** 2
**Originality:** 3
**Overall Recommendation:** 4
**Confidence:** 2

**Summary:**

This work presents the PPTEval benchmark to evaluate GUI models on PPT editing task, which contains 120 PowerPoint tasks across 12 files. PPTEval covers a wide range of editing operations such as text formatting, layout changes, image manipulation, and animations. On top of it, this work propose a hierarchical rubric-based evaluation framework to evaluate model performance by assigns partial credit and penalizes unnecessary edits, which enables scoring of intermediate progress. Experiment shows that strong computer-use agents could still struggle with these ppt editing tasks resulting much lower performance than human. Overall, this work provides a a new benchmark for measuring and advancing GUI-based computer-use agents for PPT editing tasks.

**Compliance With Llm Reviewing Policy:**

Affirmed.

**Final Justification:**

This paper explores an interesting direction of leveraging GUI agents for PPT editing. However, the current performance remains relatively limited compared to API-based approaches. Overall, I consider this work to be a weak accept.

**Key Questions For Authors:**

Please refer to the Weaknesses section above.
1. Though API method is limited in terms of functions, which tasks included in PPTEval can not be performed by API methods? Would you list some examples of it?

I would be happy to increase my score if the authors can clarify or address the concerns above in the rebuttal.

**Limitations:**

Please refer to the Weaknesses section above.

**Strengths And Weaknesses:**

Strength

1. This work addresses an practical task, PPT Editing, aligned with real-life scenarios. PPTEval is among the first to introduce a  fine-gained  benchmark for evaluating GUI agents on PowerPoint editing tasks.
2. The author’s statement about evaluation is very clear and easy to follow. And the evaluation design is dedicated and well-motivated.
3. The benchmark seems to be with high-quality. The rubric-based evaluation tree for each case is validated through human meta-evaluation, which helps ensure the reliability and quality of the benchmark.

Weakness

1. Limited human evaluation. Only two participants in human baseline, considering introduce more with different backgrounds for a more reliable result. Besides, average steps in human baseline is missing, which could be an interesting comparison.
2. Comparisons with API-based baselines. The authors argue that API-based approaches (e.g., python-pptx) are limited in terms of funcations but do not provide empirical comparisons to verify, which is important to showcase the necessity of the GUI setting of PPTEval.

---

> ### Author Rebuttal · Authors · 2026-03-31
>
> Thank you for the thoughtful review and constructive feedback. We are glad you found the PPT editing setting practically meaningful and viewed the rubric-based evaluation framework as clear, well-motivated, and high-quality. We address your concerns below.
>
> > Limited human evaluation. Only two participants in human baseline, considering introduce more with different backgrounds for a more reliable result. Besides, average steps in human baseline is missing, which could be an interesting comparison.
>
> Thanks for the feedback! We will recruit three more volunteers from different backgrounds for additional human baseline datapoints to add to the camera ready version. We could consider counting human action steps, although we had initially excluded this since it is unclear how best to measure this since even within the different agents benchmarked the action space differs (each agent was benchmarked using the action space it was trained on).
>
> > Comparisons with API-based baselines… which tasks included in PPTEval can not be performed by API methods? Would you list some examples of it?
>
> Open-source PowerPoint editing libraries like python-pptx lack apis to interact with many of the richer features of PowerPoint (e.g., certain list types, smart arts, graphics, animations and transitions, slide themes, designer, etc.) Examples of a few tasks in the benchmark that cannot done via such open-source APIs include:
> - Change list to Roman numerals (I, II, III, IV)
> - Insert a SmartArt graphic on slide 3 to represent the accounting equation (Assets = Capital + Liabilities) using a SmartArt suitable for an equation
> - Insert a black calculator icon next to the equation on slide 3
> - Apply animation effects: make the text content appear with 'Fly In' animation from the left on the references slide
> - Apply a different design theme to the entire presentation while maintaining the current color scheme of greenish backgrounds
>
> While the focus of the work is on GUI-based agents, we agree that adding an API-based baseline could be interesting, e.g., API-based agents could potentially perform better on tasks which don’t require GUI capabilities (e.g., text-based editing, tables, etc.) We will add an API-based baseline in the camera-ready version.

---

> > ### Author Rebuttal · Reviewer_1vJ7 · 2026-04-03
> >
> > Thanks for your clarification. However, the argument would be more convincing with additional experimental evidence to support it.

---

> > > ### Author Response · Authors · 2026-04-06
> > >
> > > **We have performed additional experiments to address your remaining concerns.**
> > >
> > > > Only two participants in human baseline; consider introducing more participants.
> > >
> > > During the rebuttal period, we recruited **two additional human participants**, bringing the total human baseline to **four**. One participant is a data scientist with professional experience developing presentation software, and the other is a graduate student in computer security who is familiar with PowerPoint through regular academic use.
> > >
> > > The updated human baseline is shown below:
> > >
> > > | Difficulty  |          Avg SR |       Avg Partial Score |
> > > | ----------- | --------------: | --------------: |
> > > | **Overall** | **0.80 ± 0.04** | **0.90 ± 0.03** |
> > > | Easy        |     0.88 ± 0.08 |     0.93 ± 0.06 |
> > > | Medium      |     0.79 ± 0.02 |     0.89 ± 0.03 |
> > > | Hard        |     0.67 ± 0.05 |     0.88 ± 0.07 |
> > >
> > >
> > >
> > > > Comparisons with API-based baselines...
> > >
> > > We have also evaluated an **API-based baseline**, namely Claude Code with Sonnet-4.6, which has access to open-source PowerPoint APIs such as `python-pptx` and `pptxgen-js`, along with usage instructions through Claude Code’s PPTX skill.
> > >
> > > The results are as follows:
> > >
> > > | Metric       | Overall (120) | Easy (51) | Medium (39) | Hard (30) |
> > > | ------------ | ------------: | --------: | ----------: | --------: |
> > > | Success Rate |          0.65 |      0.75 |        0.64 |      0.50 |
> > > | Avg Partial Score    |          0.83 |      0.89 |        0.84 |      0.71 |
> > >
> > > Overall, the API-based method performs better than current **GUI-based computer-use agents**, likely because it is stronger on tasks such as text edits and table manipulation, where programmatic support is available. However, it still performs **substantially worse than the human baseline**. This further highlights that there remains significant room for improvement: not only do GUI-based agents remain far from human performance, but even API-based methods do **not** saturate PPT-Eval.
> > >
> > > Furthermore, as noted in our rebuttal above, there is a **long tail of tasks** that is not currently solvable by API-based agents given the limited feature coverage of open-source PPTX libraries. In fact, we confirmed that the API-based agent receives a **score of 0** on every task listed in our original rebuttal comment. By contrast, GUI agents, despite their current limitations, are already able to solve at least some of these tasks.
> > >
> > > Examples of tasks **solved by at least one GUI agent** but **failed by the API-based agent** include:
> > >
> > > * Add a hyperlink on “HSC” linking to slide 1.
> > > * Move chart/image and caption left, and resize text to fit on the right.
> > > * Insert a SmartArt graphic on slide 3 representing the accounting equation (Assets = Capital + Liabilities) using an appropriate SmartArt style.
> > > * Switch to the “Three Content” layout and rearrange content into three columns.
> > > * Add slide numbers to all slides except the title slide.
> > > * Change a list to Roman numerals (I, II, III, IV).
> > >
> > > These failures stem from missing or limited API support for features such as **hyperlinks, SmartArt, charts, advanced layouts, equations, and advanced list/bullet formatting**.
> > >
> > > We will add a discussion of these results to the paper.
> > >
> > > With the above additional experiments, we hope we have addressed all of your concerns.

---

### Official Review · Reviewer_SEAs · 2026-03-12

**Soundness:** 2
**Presentation:** 2
**Significance:** 3
**Originality:** 3
**Overall Recommendation:** 4
**Confidence:** 3

**Summary:**

This paper introduces PPT-EVAL, a benchmark for evaluating computer-use agents on PowerPoint creation and editing tasks via the web version of PowerPoint (PowerPoint Online).

The benchmark comprises 120 tasks across 12 openly licensed PowerPoint files (404 unique slides), stratified into Easy (51 tasks), Medium (39), and Hard (30).

Tasks are executed in a sandboxed Ubuntu environment running a Chromium browser pointed to an anonymous PowerPoint Online URL, giving agents full GUI access to PowerPoint's feature set including graphics, layouts, transitions, and animations. The central contribution is a rubric-based evaluation framework: each task is represented by a tree-structured rubric with critical and non-critical nodes, a modified aggregation formula that awards partial credit while penalizing unnecessary changes (s_parent = max{0, s_crit - 0.3(1 - s_non)}), and natural-language feedback generation.

A meta-evaluation study shows Kendall's tau_b = 0.77 and Spearman's rho = 0.84 correlation with human judgments. Experiments across frontier models (Claude-4.5-Opus, Claude-4-Sonnet, Computer-Use-Preview) and open-weights models (OpenCUA, Qwen3-VL, UI-TARS) reveal that even the best model achieves only 43% success rate and 0.59 average score, far below the human baseline of 81% success rate and 0.92 average score.

**Compliance With Llm Reviewing Policy:**

Affirmed.

**Final Justification:**

Weak Accept

**Key Questions For Authors:**

1. Can you report confidence intervals for the success rates in Table 1, particularly for the Hard subset?
2. How sensitive is the rubric scoring to the choice of λ=0.3?
3. Have you considered using a different model family (e.g., GPT-4.1) for rubric evaluation to avoid potential bias when scoring Claude agents?

**Limitations:**

The authors acknowledge that rubric drafts require extensive human refinement (~150 hours), limiting scalability.

**Strengths And Weaknesses:**

Strengths

1. The rubric design is the paper's standout contribution. The tree-structured rubrics with critical/non-critical node distinction, partial credit scoring, and penalty for unintended changes solve a real evaluation challenge. The modified aggregation formula (deviating from Mind2Web 2's gate-then-average) is well-motivated by the example in Figure 6, where the original rule would yield 0 despite meaningful partial progress. The formula is simple, interpretable, and the λ=0.3 penalty parameter balances rewarding progress against penalizing extraneous edits.

2. The benchmark targets a practical and underexplored domain. PowerPoint editing is genuinely ubiquitous (the paper cites that 28.7% of business leaders spend 5+ hours per week on presentations), yet no prior benchmark tests GUI-level interaction with PowerPoint's full feature set. Table 8 (Appendix) clearly distinguishes PPT-EVAL from PPTC (API-only) and SlidesBench (generation-only).

3. The meta-evaluation validates the rubric approach rigorously.


Weaknesses

1. The benchmark scale is limited: 120 tasks across 12 files. With only 10 tasks per file and 30 hard tasks total, performance differences on hard tasks are based on very few data points.

2. The human baseline was collected from only 2 participants described as "casual rather than expert PowerPoint users." This is too thin to establish a reliable human reference. With 2 people, individual variance (familiarity with specific features, fatigue, time constraints) can substantially skew results.

3. The benchmark depends on PowerPoint Online, a commercial web service. If Microsoft changes the interface, layout, or feature availability, tasks and rubrics may break. The paper does not discuss versioning or stability strategies for long-term benchmark maintenance.

---

> ### Author Rebuttal · Authors · 2026-03-31
>
> Thank you for the thoughtful review and for recognizing the novelty of benchmarking PowerPoint-based computer-use agents, as well as the value of our rubric-based evaluation framework and thorough meta-evaluation. We address your concerns below.
>
> > The benchmark scale is limited: 120 tasks across 12 files.
>
> While our benchmark contains 120 tasks, this scale is comparable to other established computer-use agent benchmarks, including AndroidWorld [1] (116 tasks) and Mind2Web 2 [2] (130 tasks). Furthermore, with 120 tasks and even using three concurrent threads to process tasks in parallel, the benchmark already takes ~3hrs to run since computer-use agent trajectories often take several minutes to complete. Finally, we will open-source both the automated pipeline for generating rubric and task drafts as well the human instructions for refining them further to support future extension of the benchmark.
>
> [1] Rawles, Christopher, et al. "AndroidWorld: A Dynamic Benchmarking Environment for Autonomous Agents." The Thirteenth International Conference on Learning Representations.
>
> [2] Gou, Boyu, et al. "Mind2Web 2: Evaluating Agentic Search with Agent-as-a-Judge." The Thirty-ninth Annual Conference on Neural Information Processing Systems Datasets and Benchmarks Track.
>
> > Performance differences on hard tasks are based on very few data points. Can you report confidence intervals for the success rates in Table 1, particularly for the Hard subset?
>
> Yes, thanks for the suggestion. As a quick sanity check during the rebuttal period we reran the hard tasks for one model (Claude-Opus-4.5) two more times for a total of 3 runs. Its mean score on the hard task subset is 0.2778 with a std dev of 0.019, showing good stability across repeated runs. For the camera ready version, we can do additional experiments to report intervals for all of the models and all task subsets.
>
> > The human baseline was collected from only 2 participants…individual variance (familiarity with specific features, fatigue, time constraints) can substantially skew results.
>
> Thanks for the feedback! We will recruit three more volunteers from different backgrounds for additional human baseline data points to add to the camera ready version.
>
> > The benchmark depends on PowerPoint Online, a commercial web service. If Microsoft changes the interface, layout, or feature availability, tasks and rubrics may break.
>
> We provide scaffolding to run with PowerPoint Online for convenience, so that users may run the benchmark in their browsers without having to install additional software. But the benchmark is also compatible with any other application that works with pptx files (e.g., PowerPoint Desktop, Google Slides, LibreOffice Impress, or even API-based editing). As we describe in lines 151-153: “evaluation depends solely on the original file and agent’s modified version” and is agnostic of the software or method used to perform the edits.
>
> > How sensitive is the rubric scoring to the choice of λ=0.3?
>
> We set λ = 0.3 due to its intuitive meaning: penalty from performing poorly on non-critical criteria (e.g., making an accidental extraneous change or poor subjective positioning of an added textbox) at most reduces the score by 30% if all critical criteria for the task are satisfied. When developing and revising rubrics, human annotators had the λ value preset to 0.3, and so we did not experiment with different values of λ.
>
> > Have you considered using a different model family (e.g., GPT-4.1) for rubric evaluation to avoid potential bias when scoring Claude agents?
>
> Yes, to confirm that there is minimal deviation we reran evaluation for claude-sonnet-4 using gpt 4.1 on the 60 tasks that require llm/vlm judge calls. The Mean Absolute Error between using claude-sonnet-4 and gpt-4.1 for judging on these tasks is 0.1 with 78.3% of the scores lying within ±0.1. We will add these results to the appendix in the camera ready version.

---

> > ### Author Rebuttal · Reviewer_SEAs · 2026-04-02
> >
> > Thanks for the reply.

---

> > > ### Author Response · Authors · 2026-04-06
> > >
> > > Thanks for the acknowledgement. **We have now also collected additional human baseline results and hope this addresses any remaining concerns you have.**
> > >
> > > > The human baseline was collected from only 2 participants…individual variance (familiarity with specific features, fatigue, time constraints) can substantially skew results.
> > >
> > > During the rebuttal period, we recruited **two additional human participants**, bringing the total human baseline to **four**. One participant is a data scientist with professional experience developing presentation software, and the other is a graduate student in computer security who is familiar with PowerPoint through regular academic use.
> > >
> > > The updated human baseline is shown below:
> > >
> > > | Difficulty  |          Avg SR |       Avg Partial Score |
> > > | ----------- | --------------: | --------------: |
> > > | **Overall** | **0.80 ± 0.04** | **0.90 ± 0.03** |
> > > | Easy        |     0.88 ± 0.08 |     0.93 ± 0.06 |
> > > | Medium      |     0.79 ± 0.02 |     0.89 ± 0.03 |
> > > | Hard        |     0.67 ± 0.05 |     0.88 ± 0.07 |

---

### Official Review · Reviewer_UUFE · 2026-03-12

**Soundness:** 2
**Presentation:** 3
**Significance:** 2
**Originality:** 3
**Overall Recommendation:** 4
**Confidence:** 4

**Summary:**

This paper establishes a benchmark for computer-use agents on PowerPoint manipulation tasks. The authors have designed tasks of varying difficulty and a novel evaluation method. This evaluation method addresses the limitation of existing benchmarks that are often restricted to binary pass/fail assessments. The construction of the benchmark involved deep participation from both large language models and humans. According to the reported results, the evaluation method achieves high agreement with human evaluators. It is commendable that the authors have provided key details of the benchmark's construction process, rich examples in the appendix,as well as the source data and codes. The authors also evaluated both open-source and closed-source agents, and the results provide valuable guidance for assessing the capabilities of these models on PowerPoint-related tasks.

**Compliance With Llm Reviewing Policy:**

Affirmed.

**Final Justification:**

Most of the issues were addressed during the rebuttal, and I have raised my score accordingly.

**Key Questions For Authors:**

Please refer to the weaknesses.

**Limitations:**

yes

**Strengths And Weaknesses:**

Strengths：

- The paper evaluates the performance of several open-source and closed-source agents on PowerPoint manipulation tasks. The designed evaluation method is reasonable, and the reported results and analysis provide a fair understanding of the capabilities of these agents.
- The presentation of the paper is clear, and the narrative flows logically. The figures and tables are well-designed and effectively express the authors' key points, making the work easy to follow. The inclusion of abundant examples and the key prompts used in the benchmark's construction, provides exceptional transparency. This allows the reader to gain a clear and comprehensive understanding of each step in the creation process. The paper clearly explains its differences from existing research, which helps readers understand the positioning of the work within the field.
- The result of meta-evaluation indicate that the evaluation method maintains a high level of agreement with human judgment. This demonstrates the soundness of the evaluation scheme's design.
- The authors honestly acknowledge the limitations, such as the existence of LLM hallucinations.

Weaknesses:

- Regarding task diversity, while the paper claims to use PPTs from 12 different topics, my review of all 120 tasks suggests that the tasks themselves do not meaningfully differ across these topics. For instance, modifying text within a textbox is functionally the same task, whether it is in a history or an education one. Therefore, the selection of PPTs from varied backgrounds does not seem to sufficiently support the claim of the benchmark's richness and multi-dimensionality. The benchmark could be more meaningfully extended by incorporating more fundamental challenges, such as tasks involving presentations in different languages.
● I am also cautious about the significance of the tasks chosen for evaluation. The authors argue that the majority of human operations involve iterative editing rather than creation from scratch. While this may be true of human workflows, it may not reflect the primary capability users expect from an intelligent agent. The ability to generate a presentation from scratch is a more fundamental and important task. The set of tasks evaluated in this paper feel more like secondary sub-steps for refinement, modification, or beautification that would occur after an initial generation. I would support evaluating these tasks as a supplementary component to a generation benchmark, but assessing them in isolation may not fully address the core user expectation for an agent's role in presentation creation.
- The evaluation is not sufficiently thorough or in-depth. Beyond the final results, the analysis should also focus on the underlying reasons for an agent's good or poor performance. For example, is a failure caused by an error in parsing the task instruction, a problem during execution, or an incorrect understanding of the current state of the PowerPoint presentation? Although the authors provide some examples that allow readers to understand on the reasons for poor performance, this analysis should be more specific and quantitative.
- The authors report that 150 hours of human effort were required to create the evaluation rubrics for only 120 tasks. Given my observation that some of these tasks are partially redundant, this raises questions about the feasibility of extending the benchmark to a larger, more diverse, and more complex set of tasks.

---

> ### Author Rebuttal · Authors · 2026-03-31
>
> Thank you for the thoughtful feedback. We appreciate your positive assessment of the benchmark design, the evaluation methodology, the transparency of the construction process, and the breadth of models evaluated. We address your concerns below.
>
> >  Regarding task diversity, … the benchmark could be more meaningfully extended by incorporating more fundamental challenges, such as tasks involving presentations in different languages.
>
> We agree that topic diversity alone does not automatically imply task diversity. More meaningful diversity for presentation editing benchmark is the range of presentation-specific structures that the agent needs to interact with (e.g., images, textboxes, shapes, non-standard layouts, tables, etc.) We were careful to include tasks that span this diversity as shown in Fig. 4. In Fig. 7 we also show how the different model capabilities differ when attempting tasks related to these different structural elements. Fig. 5 further shows distribution of tasks across easy, medium and hard difficulties and the many high-level intent categories represented.
>
> Regarding multilingual diversity, while we agree that adding this can add another dimension of diversity, we feel that diversity related presentation-specific structures described above is more fundamental and interesting for a PPT benchmark, since multi-lingualism is not presentation-specific and other multilingual-specific benchmarks already exist to understand this capability (e.g., PangeaBench). Finally, we release all the code and guidelines we used for generating tasks and rubrics, making our benchmark extensible for studying such additional dimensions of diversity.
>
> > The authors argue that the majority of human operations involve iterative editing rather than creation from scratch. While this may be true of human workflows, it may not reflect the primary capability users expect from an intelligent agent.
>
> We disagree with the suggestion that iterative editing is less important to benchmark than creation from scratch in human-agent workflows. While from scratch presentation generation may be helpful to create an initial skeleton, given current capabilities of agents and the difference in context between agents and users, it is highly unlikely that such a draft would be accepted by the user without further iterative edits. In realistic workflows, we would still expect a majority of the time spent on improving the drafts through iterative edits, where agents can provide significant value. And as the benchmark results show, the top models currently get only a 43% success rate on such tasks (large gap with human performance of 81% SR): thus, we believe it is quite valuable to study the iterative edits setting.
>
>
> >  Beyond the final results, the analysis should also focus on the underlying reasons for an agent's good or poor performance. For example, is a failure caused by an error in parsing the task instruction, a problem during execution, or an incorrect understanding of the current state of the PowerPoint presentation?
>
> Thanks for the suggestion! We would like to clarify that none of the failures are due to problems during execution/infrastructure issues during a rollout. Any such rollouts were rerun. We are happy to add a more detailed analysis of failure reasons per model. One way to do this is to cluster the natural language reasons outputted by the rubrics into high-level patterns to get a more fine-grained understanding of the failure and success types per model. In fact, in an earlier version of the draft we had such a plot for a subset of the models, but we decided to replace it with the Fig. 7 heatmap since it provides a much more compact way to get an understanding of failures/successes by task type for all the models. We are happy to include the rubric natural language explanation clustering analysis plots for the different models in the appendix for the camera ready version.
>
> > The authors report that 150 hours of human effort were required…  this raises questions about the feasibility of extending the benchmark to a larger, more diverse, and more complex set of tasks.
>
> We agree that the 150 hours of human effort are significant, but we view this less as a weakness of the benchmark but rather an honest reflection of the effort required given current capabilities of LLMs at generating rubrics for open-ended tasks for PowerPoint. One contribution of our paper is precisely to extensively document these limitations, including the kinds of mistakes automated methods make and the amount of human intervention currently required to ensure quality (Appendix A.3 and A.5). We believe this analysis is useful to the community, especially for future work on improving automated benchmark creation. Moreover, we will open-source the task and rubric generation pipeline, which should make extension substantially easier. As LLMs improve, we expect the human effort required for scaling such benchmarks to decrease significantly.

---

> > ### Author Rebuttal · Reviewer_UUFE · 2026-04-02
> >
> > Thank you for the detailed response. While your rebuttal has clarified some aspects of the work, I find that two major concerns remain unaddressed:
> >
> > **The lack of in-depth analysis of the results.** The authors' response, stating that "in an earlier version of the draft we had such a plot..." and that "We are happy to include ... in the appendix for the camera ready version," does not provide any substantive information at this stage. A promise to add content in the final version is not a substitute for a proper rebuttal. The authors should have provided at least some illustrative examples or a summary of their preliminary findings. Since the authors stated they once had a relevant plot, I infer that supplementing this content would not require re-running experiments. In other words, considerations of time or cost are not a valid reason for its omission in the rebuttal.
> >
> > **The relationship between the 120 tasks and diversity.**  I agree that the benchmark's diversity is rooted in the variety of targeted operations and objects (text, images, tables, etc.), rather than the topics of the source files. However, the manuscript's current descriptions frequently suggest that the use of different domains is a direct contributor to task diversity, for example, "... a benchmark of 120 diverse PowerPoint tasks ...", "PPT-EVAL comprises 120 high-quality tasks sourced from diverse, openly licensed PowerPoint decks ...". This framing is misleading. The authors should clarify this by emphasizing that the diversity is in the tasks themselves, and frame the different PPT topics as merely the "testing substrate", rather than a component of the benchmark's diversity.

---

> > > ### Author Response · Authors · 2026-04-07
> > >
> > > Thanks for the follow-ups.
> > >
> > > > In-depth analysis of results
> > >
> > > We reran the clustering analysis we had done on a subset for all the models now. Here is a link to the figure: https://anonymous.4open.science/r/ppt-eval-figures-72DE/success_failure_analysis.pdf. The first heatmap in the figure (left) clusters the tasks into high-level intent categories and then plots the success rate of the different models for each of the intent categories. The second heatmap (right) clusters the natural language explanations provided by the rubrics for failed tasks across models into common failure patterns, and then plots the percentage of failures of each model corresponding to each of these patterns. We will include this analysis in the paper appendix along with a discussion. We would also like to remind the reviewer that we also provide an existing breakdown of results by the object type relevant to the task in Fig. 7, a discussion of these results in Section 5.1 and qualitivate examples of trajectories in Appendix E.
> > >
> > >
> > > > The relationship between the 120 tasks and diversity.
> > >
> > > Thank you for the feedback on the phrasing. We will rephrase the text to make it clear that the primary source of meaningful diversity in this benchmark largely stems from the different kinds of objects (text, images, tables, etc.) that the tasks require the agent to interact with, rather than the topics of the files or the languages used.
> > >
> > > We hope this helps address your remaining concerns.

---

### Decision · Program_Chairs · 2026-04-30

**Decision:**

Accept (regular)

**Comment:**

This paper introduces a benchmark for computer-use agents called PPT-Eval, that consists of 120 PowerPoint tasks. To measure progress on this benchmark, the paper proposes an evaluation framework that creates task-specific rubrics to provide partial credit based on intermediate steps. The authors show that their rubric-based evaluation correlates well with human judgements. The paper evaluates both frontier models (Claude-4.5-Opus, Clause-4-Sonnet, Computer-Use-Preview) and open-weight models (OpenCUA, Qwen3-VL, UI-TARS). The authors find that even strong models perform relatively poorly on the proposed PowerPoint tasks, leaving significant room for improvement.

The reviewers found that the paper addresses an underexplored and realistic problem of PowerPoint editing using GUI rather than API interaction, which makes the task more challenging. They found the tree-structured rubric design to be a well thought-out and significant contribution, and the aggregation formula for partial credit to be well motivated. They also found the paper to be well-written with clear figures and tables, and appreciated the meta-evaluation of the rubric-based approach, showing good agreement with human judgement. They found that the results and analysis provide a good understanding of the capabilities of these agents.

Reviewers raised concerns regarding the scale of the benchmark, containing only 120 tasks using 12 files. The authors’ rebuttal addressed this concern by noting that other computer-use benchmarks like AndroidWorld and Mind2Web 2 have similar sizes (116 tasks and 130 tasks, respectively).

Reviewers also raised concerns regarding the task diversity, the human effort involved in creating rubrics, the lack of comparisons to API-based baselines rather than GUI-based ones, and the fact that the human baseline used only two casual PPT users. These concerns were mostly addressed by the rebuttal.

Reviewer UUFE also pointed out that the tasks in the paper focus on iterative edits rather than PPT generation from scratch, and that the evaluation lacks a detailed failure analysis. I agree with the authors that iterative editing is an interesting and important capability to evaluate.

After the rebuttal, Reviewer 1vJ7 noted that using GUI-based agents for PPT editing is interesting, but that their performance is limited compared to API-based approaches. However, I think this still illustrates the importance of the PPT-Eval benchmark, as the community could benefit from such GUI-based benchmarks.

Overall, the reviewers voted unanimously for acceptance. Concerns regarding the scale of the benchmark and task diversity were sufficiently addressed by the rebuttal.

This paper presents a realistic benchmark for PowerPoint editing based on the GUI interface, that shows that there exist capability gaps for current agents, and will be of interest to the ICML community.